



# Impact of non-normal flow rule on linear kinematic features in pan-Arctic ice-ocean simulations

Jean-François Lemieux[1], Mathieu Plante[1], Nils Hutter[2], Damien Ringeisen[3], Bruno Tremblay[4], François Roy[1], and Philippe Blain[5]

[1]Recherche en Prévision Numérique Environnementale/Environnement et Changement Climatique Canada, 2121 route Transcanadienne, Dorval QC, Canada.
[2]GEOMAR Helmholtz-Center for Ocean Research Kiel, Kiel, Germany.
[3]Canadian Centre for Climate Modelling and Analysis, Climate Research Division, Environment and Climate Change Canada, Victoria, British Columbia, Canada.
[4]Department of Atmospheric and Oceanic Sciences, McGill University, Montréal QC, Canada.
[5]Service Météorologique Canadien, Environnement et Changement Climatique Canada, 2121 route Transcanadienne, Dorval QC, Canada.

**Correspondence:** Jean-François Lemieux (jean-francois.lemieux@ec.gc.ca)

**Abstract.**

One important conclusion of the Sea Ice Rheology Experiment (SIREx) is that continuum based sea ice models, with different spatial discretizations and/or sea ice rheologies, all simulate intersection angles between linear kinematic features (LKFs) that are too wide compared to observations. The peak of the probability density function (PDF) of simulated intersection angles is around 90° while the PDF for observed angles rather exhibits a peak around 45°. Ringeisen et al. (2021) proposed to remedy this issue for viscous-plastic (VP) and elastic-VP (EVP) models by introducing a non-normal flow rule specified by a plastic potential. We implemented the plastic potential approach of Ringeisen et al. (2021) in the CICE sea ice model. In pan-Arctic simulations, the non-normal flow rule also leads to a peak of the PDF around 90°. We show that this peak at 90° is at least partly a consequence of many LKFs that are aligned with the computational grid. Nevertheless, the non-normal flow rule brings an interesting capability: it could be used to independently optimize simulated LKFs and more generally deformations while parameters defining the yield curve could serve for modifying simulated landfast ice and to a lesser extent sea ice drift.

## 1 Introduction

The ability of large-scale continuum-based sea ice models to simulate linear kinematic features (LKFs) and more generally deformations was evaluated in the Sea Ice Rheology Experiment (SIREx, Bouchat et al., 2022; Hutter et al., 2022). Various methods and metrics were used to quantify the quality of simulated deformations. These included PDF of deformations, temporal and spatial scaling, LKF density, length of LKFs, intersection angles and LKF lifetime. For most metrics, there were notable differences in the quality of the simulations with some models that performed well while others struggled to properly represent the deformations. However, all the models, whatever the rheology, failed to correctly represent the distribution of



intersection angles between conjugate pairs of LKFs (i.e., LKFs that form simultaneously under compressive stresses). Indeed, the peak of the distribution is around 45° for SAR derived observed deformations (as recently confirmed, Ringeisen et al., 2023) while the models lead to simulated intersection angles with a peak of the PDF around 90° (Hutter et al., 2022).

Prior to SIREx, there were already a few indications that viscous-plastic (VP) models tend to simulate too large intersection angles. Ringeisen et al. (2019) demonstrated in uni-axial loading experiments that the standard VP rheology cannot simulate fracture angles smaller than 60° (i.e., $2 \times 30°$). Hutter and Losch (2020) analyzed high-resolution MITgcm pan-Arctic simulations and found that intersection angles are too wide compared to observations with a peak of the distribution around 90°.

The yield curve and the flow rule are two important characteristics to define a VP rheology. The yield curve (e.g. in stress invariant space) specifies the critical stresses associated with the failure of sea ice in shear/compression or shear/tension. The flow rule defines the post-failure deformations. It is expressed with the use of a plastic potential: the post-failure deformations are normal to the plastic potential.

The standard VP rheology of Hibler (1979) is based on an elliptical yield curve and a normal flow rule. The normal flow rule implies that the plastic potential is the same as the yield curve and that the post-failure deformations are normal to the yield curve. Ringeisen et al. (2021) attributed to the normal flow rule the inability of the VP model to simulate small intersection angles. These authors proposed to define the plastic potential independently of the yield curve, therefore allowing a non-normal flow rule. Uni-axial idealized compression experiments with a non-normal flow rule demonstrated that smaller intersection angles, more in line with observations, can be obtained and that the intersection angles of LKFs can be precisely linked to the shape of the yield curve and plastic potential following the Arthur angles (Arthur et al., 1977).

We have implemented, in the CICE sea ice model, the plastic potential approach of Ringeisen et al. (2021). In this work, using more realistic pan-Arctic simulations, we investigate the impact of a VP rheology with a non-normal flow rule on simulated LKFs and especially on intersection angles. The goal of this study is not to do a thorough assessment of the realism of simulated LKFs against observations but rather to conduct a sensitivity study of the impact of non-normal flow rules in simulating LKFs and more generally on the simulated sea ice cover.

We will answer the following questions: 1) what is the impact of non-normal flow rules on statistics of LKFs such as length, number, density and especially intersection angles? 2) are smaller intersection angles simulated in pan-Arctic experiments when using a non-normal flow rule?

This paper is structured as follow. A brief presentation of the plastic potential and the flow rule is given in section 2. Details about the experimental setup and the methodology are respectively given in sections 3 and 4. Results are presented and anal-





ysed in section 5. Broader implications are discussed in section 6 while concluding remarks are provided in section 7.


## 2   The plastic potential and the flow rule

This section provides a short overview of the non-normal flow rule and of the plastic potential for the VP rheology with an elliptical yield curve. More details can be found in Ringeisen et al. (2021).

The yield curve in the VP rheology defines the critical stresses associated with plastic failure. To close the system of equations, assumptions have to be made about the post-failure deformations, that is the flow rule. The flow rule can be formulated with the use of a plastic potential. The idea is that post-failure deformations are normal to the plastic potential.

In the original VP rheology of Hibler (1979), the plastic potential is the same as the elliptical yield curve. This leads to a
normal flow rule: the post-failure deformations are normal to the yield curve. The constitutive equation can be expressed using $\zeta$ and $\eta$ which are respectively the bulk and shear viscosities (Hibler, 1979). Given the ellipse ratio $e$ of major to minor axes, $\eta = e^{-2}\zeta$, with $\zeta$ expressed as

$$\zeta = \frac{P(1+k_t)}{2\Delta},\tag{1}$$

where $P$ is the ice strength, $k_t$ is a parameter between 0 and 1 that determines tensile strength (König Beatty and Holland, 2010)
and $\Delta = \left[ D_d^2 + e^{-2}(D_t^2 + D_s^2) \right]^{1/2}$ with $D_d = \dot{\epsilon}_{11} + \dot{\epsilon}_{22}$ the divergence, $D_t = \dot{\epsilon}_{11} - \dot{\epsilon}_{22}$ the horizontal tension and $D_s = 2\dot{\epsilon}_{12}$ the shearing strain rate. These deformations are defined from the components $\dot{\epsilon}_{11}$, $\dot{\epsilon}_{22}$ and $\dot{\epsilon}_{12}$ of a symmetric strain rate tensor.

Using an elliptical yield curve, Ringeisen et al. (2021) implemented a modified VP rheology that relies on another ellipse defining the plastic potential. Following the notation of Ringeisen et al. (2021), $e_{\mathrm{F}}$ defines the aspect ratio of the yield curve
while $e_{\mathrm{G}}$ defines the one of the elliptical plastic potential. Setting $e_{\mathrm{G}} = e_{\mathrm{F}} = e = 2$, one recovers the VP rheology introduced by Hibler (1979).

With the non-normal flow rule, $\eta$ is now equal to $e_{\mathrm{G}}^{-2}\zeta$. The bulk viscosity $\zeta$ is in this case expressed as in Eq. (1) above but with a slight modification to the definition of $\Delta$, that is

$$\Delta = \left[ D_d^2 + \frac{e_{\mathrm{F}}^2}{e_{\mathrm{G}}^4}(D_t^2 + D_s^2) \right]^{1/2}.\tag{2}$$

Note that there is a sign that is incorrect in the definition of $\Delta$ in Ringeisen et al. (2021, their Eq.15).





When the deformation $\Delta$ tends toward zero, $\zeta$ tends toward infinity. To prevent this singularity, $\Delta$ is replaced by $\Delta^*$ in Eq. (1). In the simulations presented in this article, $\Delta^*$ is defined by the capping approach of Hibler (1979), that is

$\Delta^* = \max(\Delta, \Delta_{\min})$, where $\Delta_{\min}$ is a small deformation set here to $10^{-11}\mathrm{s}^{-1}$.

## 3   Experimental setup

Our experimental setup is based on the CICEv6.5 sea ice model (Hunke et al., 2023) and the NEMOv3.6 ocean model (Madec, 2008). We use a regional configuration with a domain that covers the Arctic Ocean, the oceanic regions around Canada, the

North Atlantic and the North Pacific. This setup was developed as part of the Canadian Operational Network of Coupled Environmental Prediction Systems (CONCEPTS) initiative. The regional grid used for the simulations is the CONCEPTS REGional 1/12° (CREG12) grid which is used in the Environment and Climate Change Canada (ECCC) operational system RIOPS (Smith et al., 2021).

The ocean model is applied in a variable volume and nonlinear free surface formulation, using seventy-five vertical levels. Ocean mixing is parameterized with the NEMO k-epsilon approach. Oceanic boundary conditions for the North Pacific and North Atlantic are taken from the GLORYS2 version 4 reanalysis (Garric et al., 2017). Monthly averages of vertical profiles of ocean currents, temperature, and salinity are applied at the boundaries. A time-splitting technique with a sub–time step of 5 s is used for the treatment of the nonlinear free surface (including the tides). For the tides, vertically averaged velocities (13

harmonic components) were obtained from the Oregon State University tidal prediction model. At the open boundaries, the barotropic part of the velocity components are prescribed following the method of Flather (1976). Sea surface height forcing includes the tidal potential, self-attraction and loading effects.

Ten sea ice thickness categories, as defined in Smith et al. (2016), are used for the simulations. The thermodynamics com-

ponent uses the approach of Bitz and Lipscomb (1999). The parameterization of Hibler (1979) is employed for the formulation of the ice strength. Grounding of ice keels in shallow water is parameterized with the scheme of Lemieux et al. (2016).

The ice-ocean simulations are forced by 33-km resolution ECCC atmospheric reforecasts (Smith et al., 2014). As the ECCC atmospheric forcing dataset only covers the period from 2001 to 2010, this puts some constraints on the spinup and analysis

period. The analysis period is 25 September 2004 to 31 May 2008. The restart on 25 September 2004 was obtained following the approach described in Lemieux et al. (2018). It is obtained from a CICEv4-NEMOv3.6 simulation consisting of a pseudo-spinup followed by a final spinup.

The pseudo-spinup is initialized with average (September – October 2001) sea ice concentration from the National Snow

and Ice Data Center (NSIDC, https://nsidc.org/data/seaice_index/) and the average (October-November 2003) sea ice thickness




| Symbol | Definition | spinup | exp |
|---|---|---|---|
| $P^*$ | ice strength parameter | 27.5 kNm$^{-1}$ | 22.5 kNm$^{-1}$ |
| $C^*$ | concentration strength parameter | 15 | 15 |
| $k_t$ | isotropic tensile strength parameter | 0.05 | 0.00 |
| $z_{0ai}$ | air-ice roughness | 5.7×10$^{-4}$ m | 5.4×10$^{-4}$ m |
| $z_{0oi}$ | ocean-ice roughness | 0.0182 m | 0.0180 m |

**Table 1.** Relevant dynamical parameters for the spinup (and pseudo-spinup) and for the numerical experiments

| | $e_F$ | $e_G$ |
|---|---|---|
| exp1 | 2.0 | 3.0 |
| exp2 | 2.0 | 2.0 |
| exp3 | 2.0 | 1.33 |
| exp4 | 1.75 | 1.16 |
| exp5 | 1.75 | 1.75 |
| exp6 | 1.75 | 2.63 |
| exp7 | 1.5 | 2.25 |
| exp8 | 1.5 | 1.5 |
| exp9 | 1.5 | 1.0 |
| exp10 | 1.16 | 1.75 |
| exp11 | 2.63 | 1.75 |

**Table 2.** List of numerical experiments with values of $e_F$ and $e_G$.

field from ICESat data (https://nsidc.org/data/icesat). The initial temperature and salinity for the ocean are averages (September–October) of WOA13_95A4 fields (Locarnini et al., 2013; Zweng et al., 2013). The ocean starts at rest; the sea surface height field and ocean currents are set to zero. The pseudo-spinup consisted in running the coupled model three times from 1 October 2001 to 30 September 2002.


The pseudo-spinup solution was then used to start the spinup again on 1 October 2001. CICEv4-NEMOv3.6 then ran up to 25 September 2004. The solution on 25 September 2004 is the restart that is used for the simulations. The sea ice part of the restart was converted in order to be used by CICEv6.5.

For the pseudo-spinup and the spinup, we use the 'optimal' parameters of Chikhar et al. (2019). Table 1 lists the values of the most relevant parameters for the spinup procedure, that is the ones associated with rheology and sea ice dynamics (note that $e$=1.5 for the pseudo-spinup and spinup). The advective time step is 180 s for the spinup procedure and for all the simulations.



Following the advice of Bouchat et al. (2022), a large number (900) of EVP subcycles were performed in order to ensure numerical convergence of the solutions.


## 4 Methodology

### 4.1 Sensitivity analysis

From the ice-ocean restart described in section 3, we ran a series of numerical experiments with different values of $e_F$ and $e_G$. To simulate smaller intersection angles, Ringeisen et al. (2021) suggest to set $e_G < e_F$. Nevertheless, as this is a sensitivity
study, we also test for values of $e_G$ larger than $e_F$ keeping in mind that the most important results are for $e_G < e_F$. For each value of $e_F$, $e_G$ is set to $\sim e_F/1.5$ or $e_F$ or $\sim 1.5e_F$ (these are referred to experiments 1 to 9). As $e_F = 1.75$ is between the two other values tested (i.e., $e_F = 1.5$ and $e_F = 2.0$), we sometimes only show results associated with this value when the conclusions are qualitatively the same for the other values of $e_F$. We add two other experiments to see the impact of $e_F$ for a constant $e_G$ of 1.75 (referred to as experiments 10 and 11). The values of $e_F$ and $e_G$ for the 11 numerical experiments are
given in table 2. Except for $C^* = 15$, the other dynamical parameters for the experiments are slightly different than the ones used for the spinup procedure. $P^*$, $z_{0ai}$ and $z_{0oi}$ are set to the values of the latest version of RIOPS (Smith et al., 2021). As in Ringeisen et al. (2021), $k_t$ is set to zero. The last column of Table 1 provides the values of these parameters. Again, the number of EVP subcycles is set to 900.

These experiments cover the period 25 September 2004 to 31 May 2008. We focus on the formation of LKFs when the sea ice is compact in the Arctic Ocean. We therefore analyse the LKFs between January 1 to May 31 for the years 2005 to 2008. We focus our analysis on the first winter-spring period (i.e., 2005) because the sea ice thickness fields of the different simulations are more similar. Nevertheless, we also analyze LKF statistics for the other winter-spring periods to verify that the same conclusions apply. We use 00 UTC snapshots of sea ice deformations to investigate the simulated LKFs. Daily mean
of the sea ice volume are also stored to study the impact of the non-normal flow rule on total sea ice volume in the northern hemisphere.

### 4.2 LKF detection and analysis

To detect LKFs, we downloaded version 2.0 of Nils Hutter's LKF package (https://github.com/nhutter/lkf_tools/releases/tag/
v2.0, Hutter et al. (2019)) and made a few minor modifications for our CICE outputs. A kernel value of seven is used for the detection algorithm. LKFs are detected only if the sea ice concentration is larger than 0.15 and if they are located in a region in the central Arctic defined by a mask. We refer to this mask as the pack ice mask. It covers the region shown in gray in Fig.2. It corresponds to all the CREG12 grid cells in the Arctic Ocean that are at least 250 km away from the nearest land cell. This





mask ensures that LKFs are detected and analysed in compact ice, away from the coasts and from regions of landfast ice (i.e.
excluding coastal or flaw polynyi).

Our analysis focuses on spatial characteristics of LKFs and leave aside temporal aspects such as the lifetime of LKFs. To
analyse the detected LKFs, we developed a set of tools written in Python. These tools can be obtained on GitHub (https:
//github.com/JFLemieux73/lkf_tools). Many of the metrics used to analyse the LKFs are the same ones introduced in Hutter
and Losch (2020) and Hutter et al. (2022). However, we introduce two new metrics: the width of LKFs and the angle with the
grid.

The detection algorithm uses a morphological thinning method that reduces the width of LKFs to one pixel (Hutter et al.,
2019). In the most recent implementation a point of the thinned LKFs correspond to the largest total deformation along a
perpendicular transect (Hutter, 2023). Given the location of a LKF point with a total deformation of $\dot{\epsilon}_{tot}$, our new Python
algorithm for the width calculates the number of pixels (i.e. grid cells) required in both perpendicular search directions for
reducing the total deformation below $\alpha\dot{\epsilon}_{tot}$ where $0 < \alpha < 1$. The parameter $\alpha$ is set to 0.5 in this study. More details about the
LKF width algorithm are provided in Appendix A.

Our LKF analysis tool also estimates angles of LKFs with the computational grid at the intersection points of conjugate
fault lines. As such, it first identifies pairs of intersecting LKFs. Following Hutter et al. (2022) and Ringeisen et al. (2023), the
vorticity of both intersecting LKFs is analysed. If the two LKFs have opposite signs of vorticity, the intersecting pair can be
interpreted as conjugate LKFs. As shown in Fig. B1 in Hutter et al. (2022), the stress direction can be determined from the
vorticity field; the intersection angle obtained is between 0 and 180°. For each intersecting LKF, our Python tool then calculates
the acute angles $\theta_x$ and $\theta_y$ between a LKF and the (local) $x$ and $y$ axis of the computational grid. The metric that we use is the
minimum angle with the grid, that is $\theta_{min} = \min(\theta_x, \theta_y)$. More details about our algorithm for calculating intersection angles
and angles with the grid are given in Appendix B.

### 4.3 Spatial scaling

As a complement to the LKF metrics described above, we investigate the spatial scaling response to the plastic potential. This
metrics is not a direct measure of the organization of sea ice deformations into LKFs but measures the level of spatial local-
ization of the simulated deformations. While not as intuitive as the LKF characteristics, this metric has the benefit of being
independent from the LKF detection methods, thus generalizing the results.

The spatial scaling is performed by applying the same coarse-graining methods as described in Bouchat and Tremblay
(2020) and Bouchat et al. (2022), but here applied to Eulerian sea ice deformations (i.e., without the initial step of computing
Lagrangian trajectories, Hutter et al. (2018)). That is, for a given coarsened scale $L$, the mean sea ice deformation is computed





first by aggregating to scale the Finite Difference sea ice velocity gradients (using an area-weighted average), then computing the field-average of these coarse-grained deformations. This method is repeated across multiple scales (multiples of the nominal resolution), and the spatial scaling is defined by the slope $\beta$ of the regression line in logarithmic space that best represents the results.

Following Bouchat and Tremblay (2020) and Bouchat et al. (2022), the aggregated sea ice deformations are further weighted by their signal-to-noise ratio. This method was designed to apply similar post-processing methods to model data as those applied to satellite observations, with the added benefit of presenting an enhanced sensitivity to the level of feature localization. This method is adapted to our Eulerian approach by setting the tracking error to zero and the sea ice velocity error to the model precision, yielding a unique noise value for all deformations. The signal-to-noise weighting method therefore corresponds to weighting the sea ice deformations by their own magnitude.

## 4.4 Ratio of divergence and shearing

Because the post-failure deformations are normal to the plastic potential, a simulation with a large value of $e_G$ should exhibit less convergence/divergence and more shear than a simulation with a smaller value of $e_G$. To investigate the impact of the plastic potential on the post-failure deformations, the amount of divergence compared to shear is quantified by calculating

$$\theta_r = \tan^{-1}\left(\frac{\dot{\epsilon}_{\mathrm{I}}}{\dot{\epsilon}_{\mathrm{II}}}\right), \tag{3}$$

where $\dot{\epsilon}_{\mathrm{I}} = D_d$ and $\dot{\epsilon}_{\mathrm{II}} = (D_t^2 + D_s^2)^{1/2}$ are the strain rate invariants. The angle $\theta_r$ varies between -90° (pure convergence) to 90° (pure divergence) with pure shear deformation corresponding to $\theta_r = 0°$.

This metric is calculated from 00 UTC snapshots $\dot{\epsilon}_{\mathrm{I}}$ and $\dot{\epsilon}_{\mathrm{II}}$. Note that $\dot{\epsilon}_{\mathrm{I}}$ and $\dot{\epsilon}_{\mathrm{II}}$ are defined at the T-point (i.e., cell center) in the CICE outputs.

## 5 Results

### 5.1 Impact of non-normal flow rule on simulated LKFs

Snapshots of the total deformation show that $e_G$ has a strong impact of the definition of LKFs. This is shown in Fig. 1 for 25 April 2005 for which $e_F$ is kept constant at 1.75. For $e_G = 2.63$ (Fig. 1a), LKFs are blurry and not well defined. LKFs are better defined as $e_G$ is reduced (Fig. 1b and c). As $e_G$ is reduced, the algorithm detects more LKFs for these very same snapshots (Fig. 2).





Visually speaking, Figs 1 show that similar large-scale patterns of deformation are present for different values of $e_G$. However, these panels indicate that a smaller $e_G$ leads to narrower LKFs. To verify this, we use our novel Python tool for calculating
the width of detected LKFs (see Appendix A for details). The mean width of detected LKFs in the pack ice region over the period 1 January to 31 may 2005 is indeed sensitive to the value of $e_G$ (see Fig. 3). For all the values tested of $e_F$, a non-normal flow rule with $e_G < e_F$ leads to narrower LKFs compared to the normal flow rule (i.e., $e_G = e_F$).

Because LKFs are blurry for large values of $e_G$ (Fig. 1a), the number of detected LKFs is smaller. This is shown in Fig. 5
for $e_F = 1.75$. The mean number of detected LKFs over the period 1 January to 31 may 2005 is 63.2 for $e_G = 2.63$, 86.2 for $e_G = 1.75$ and 106.5 for $e_G = 1.16$. These conclusions also remain valid for the other values of $e_F$ (and associated $e_G$) tested and for the other winter-spring periods (2006-2008) of the simulations (not shown).

Detected LKFs also tend to be longer for a smaller $e_G$. This is clearly displayed in Fig. 6. The increase in the mean length is
more pronounced going from $e_G$=2.63 to $e_G$=1.75 than from $e_G$=1.75 to $e_G$=1.16. The length of a detected LKF is obtained by summing the distances between two neighboring points. These distances are calculated with the haversine formula. The mean of the mean length over the period 1 January to 31 may 2005 is respectively 167.4 km, 221.7 km and 257.4 km for $e_G$=2.63, $e_G$=1.75 and $e_G$=1.16. These conclusions remain valid for the other values of $e_F$ (and associated $e_G$) tested and for the other winter-spring periods (2006-2008) of the simulations (not shown). Because a smaller $e_G$ leads to longer and more numerous
LKFs, the total length of LKFs is also larger (see Fig. 7). The mean total length over the period 1 January to 31 may 2005 is respectively 10739 km, 18746 km and 26546 km for $e_G$=2.63, $e_G$=1.75 and $e_G$=1.16.

The more numerous simulated LKFs when decreasing the value of $e_G$ is an ubiquitous phenomenon in the region defined by our pack ice mask as demonstrated by the LKF density maps in Fig. 8. Given the $n_s = 151$ snapshots (00 UTC) for the period
1 January to 31 May 2005 that were used for the LKF detection, the density is calculated as the number of incidences a pixel is crossed by a LFK divided by $n_s$. The density clearly increases as $e_G$ is reduced. These conclusions remain the same for the other values of $e_F$ tested (not shown).

One limitation of the LKF metrics discussed above is that they do not consider the smoother or shorter features excluded by
the LKF detection algorithm (see for instance the features in Fig. 1a in the region of the pack ice mask vs. the LKFs detected in Fig. 2a). As such, these metrics are likely unable to capture the model sensitivity at high $e_G$ values. This might explain the ambiguous impact of $e_G > e_F$ on the mean LKF width (Fig. 3). To generalize the results and characterize the level of localization in the full sea ice deformation fields, we additionally calculate the spatial scaling from the nominal ($\sim$4 km) to $\sim$500 km scales (Fig. 4). This exercise confirms that the level of localization in the simulated sea ice deformation fields is sensitive to $e_F$
but mostly sensitive to the $e_G$ parameter, with increased scaling (larger scaling exponent $\beta$) for lower $e_G$.





## 5.2 Impact of non-normal flow rule on intersection angle

In idealized uniaxial compressive experiments, Ringeisen et al. (2021) demonstrated that, compared to $e_\mathrm{G} = e_\mathrm{F}$ (the normal flow rule case), a plastic potential with $e_\mathrm{G} < e_\mathrm{F}$ leads to smaller intersection angles between conjugate fault lines (or conjugate

LKFs) for $e_\mathrm{F} > 1$. Here, we investigate the impact of non-normal flow rule on the intersection angles between conjugate LKFs in pan-Arctic simulations. Details about our Python algorithm for calculating the angle between conjugate LKFs (and angles with the grid) are provided in Appendix B.

Fig. 9b shows the PDF of intersection angles for conjugate pairs (i.e., $\theta_c$), for the period 1 January to 31 May 2005, for

$e_\mathrm{G} = e_\mathrm{F} = 1.75$. Note that short LKFs are excluded from this analysis; the intersection is not taken into account in the PDF if at least one of the two intersecting LKFs has less than $N_{min} = 10$ points. The peak of the distribution is at 90°. This is unrealistic compared to observations of intersection angles (Hutter and Losch, 2020; Hutter et al., 2022; Ringeisen et al., 2023). Reducing (Fig. 9c) or increasing (Fig. 9a) $e_\mathrm{G}$ does not improve the PDF. The PDFs in Fig. 9 are similar to the ones obtained by Hutter et al. (2022) for SIREx.


Contrary to the results of Ringeisen et al. (2021) for idealized uni-axial compression experiments, the angles between conjugate fault lines are not reduced when setting $e_\mathrm{G} < e_\mathrm{F}$ in our pan-Arctic simulations (see Fig. 9). Note that Ringeisen et al. (2021) produced these idealized experiments with an implicit solver; in other words they obtained a VP solution as opposed to our EVP results presented here. To verify that the too large intersection angles obtained here are not caused by the EVP for

$e_\mathrm{G} < e_\mathrm{F}$, we made use of a new capability of CICE: a Picard implicit solver similar to the one developed by Lemieux et al. (2008). The VP approach in CICE is based on the same spatial discretization on the B-grid as the EVP (Hunke and Dukowicz, 2002). Using 10 nonlinear iterations for the implicit solver, we ran experiments for $e_\mathrm{G} = e_\mathrm{F} = 1.75$, $e_\mathrm{G} = 1.16, e_\mathrm{F} = 1.75$, and $e_\mathrm{G} = 2.63, e_\mathrm{F} = 1.75$. PDFs of the intersection angle of conjugate pairs resemble the ones obtained with the EVP: the peaks are also at 90° for the three experiments (not shown).


As the peaks of all these distributions are close to 90°, one might consider that many LKFs are aligned with the computational grid. Hutter and Losch (2020) briefly investigated this in their MITgcm simulations by comparing the mean orientation of LKFs with grid orientation and could not find a clear correlation. Hutter et al. (2022) also argued that the peak at 90° is not due to LKFs aligned with the grid because SIREx simulations from unstructured grid models also exhibit a peak of the PDF

close to 90°.

However, a visual inspection of simulated deformation fields produced by our CICE B-grid model (such as the ones shown in Fig. 1) indicates that many LKFs are aligned with the grid. To investigate this more objectively, we use our novel Python tool that calculates the angles of LKFs with the (local) $x$ and $y$ axis of the grid. The angles $\theta_x$ and $\theta_y$ with the $x$ and $y$ axis are calculated at the intersection point of conjugate pairs. Fig. 10 shows the PDFs of the minimum angle $\theta_{min} = \min(\theta_x, \theta_y)$





of all the conjugate LKFs for the period 1 January to 31 May 2005. Given $e_F = 1.75$, the peak of the PDF is clearly at $0°$ for the three values of $e_G$ tested (Fig. 10). Although decreasing $e_G$ slightly modifies the PDF toward more uniform values, there is still clearly a tendency for LKFs to align (or nearly align) with the computational grid. Due to the continents and large-scale sea ice drift patterns, the same PDF calculated using observed LKFs would not necessarily be uniform.


We finally verified whether the peak of the PDF of $\theta_c$ at $90°$ and of the one of $\theta_{min}$ at $0°$ are not caused by short LKFs that would align more easily with the grid. Increasing $N_{min}$ from 10 to 20 or 30 (i.e., analysing only longer and longer LKFs) does not change, qualitatively, our conclusions (not shown). These conclusions remain valid for the other values of $e_F$ tested (not shown).


## 6 Broader implications and discussion

Results from section 5 show there is an increase in LKF activity when reducing $e_G$ for a fixed value of $e_F$ (see for example Fig. 7 and Fig. 8 ). This suggests this could have an impact on the sea ice formation in leads.

To verify this, we calculated the total sea ice volume in the Northern hemisphere, as a function of time, for different values of $e_F$ and $e_G$. Fig. 11a shows the total simulated volume for a fixed $e_F = 1.75$ for different values of $e_G$. Compared to the Pan-Arctic Ice Ocean Modeling and Assimilation System (PIOMAS, Schweiger et al., 2011), the simulations already start on 25 September 2004 with a thicker sea ice cover. The impact of $e_G$ is striking; smaller values of $e_G$ lead to a thicker sea ice cover. We anticipate that the impact of $e_G$ on the sea ice volume would be smaller in a coupled configuration with an 310    atmospheric model; the forced setup probably causes an overestimation of the sea ice growth as feedbacks between the ocean and the atmosphere are not represented. On the other hand, Fig. 11b demonstrates that the impact of $e_F$ on the total volume is negligible compared to the one of $e_G$.

We interpret this effect of $e_G$ on the sea ice volume to the amount of divergence associated with the flow rule: a smaller 315    $e_G$ leads to a more circular (or less elliptic) plastic potential and therefore normal vectors exhibiting more divergence. This enhanced lead formation with a smaller $e_G$ should be associated with an increased in the sea ice growth. We also argue that the increased divergence associated with the non-normal flow rule ($e_G < e_F$) also explains the higher LKF density (see Fig. 8) and the reduced mean width (see Fig. 3) of simulated LKFs compared to results with the normal flow rule ($e_G = e_F$). More divergence decreases the ice strength, concentrates spatially the LKFs and favors their persistence.


To quantify the amount of divergence ($\dot{\epsilon}_I$) compared to shear ($\dot{\epsilon}_{II}$), we use the angle $\theta_r$ as introduced in Eq. (3). Given daily snapshots at 00 UTC for the period 1 January to 31 May 2005, $\theta_r$ is calculated at all the ocean grid cells inside the pack ice mask when $\Delta > \Delta_{min}$ (i.e., the state of stress is plastic). Fig. 12a shows the PDF of $\theta_r$ for different $e_G$ (with a fixed $e_F$).





Similarly, Fig. 12b shows the PDF of $\theta_r$ for different $e_F$ (with a fixed $e_G$). $e_G$ has a strong impact on the amount of divergence
compared to shear while the impact of $e_F$ is small when keeping $e_G$ fixed. The PDF for a large value of $e_G$ exhibits many pure
shear events (i.e. $\theta_r = 0$); decreasing $e_G$ augments the events of shear plus divergence (or convergence). For all PDFs, the pure
divergence or pure convergence events are rare.

Because the sea ice volume time series diverge more and more with time we only look at the impact of $e_F$ and $e_G$ on the
sea ice drift for the period 1 January to 31 may 2005. We choose this time interval because it is close to the beginning of
our simulations and during a period (winter-spring) for which rheology has a strong impact on the drift. We interpolated daily
averaged velocity components from the U-point to the T-point where concentration is calculated. Using these, we calculated
the daily spatial averaged sea ice speed. This spatial average is computed for all the grid cells with a concentration larger than
0.15 inside the pack ice mask. Monthly mean values of the daily spatial averaged sea ice speed are then calculated. The results
are shown in Fig. 13.

The impact on the drift of changing the size of the yield curve (i.e., $e_F$) is more important than changing the flow rule (i.e.,
$e_G$). In fact, compared to the normal flow rule ($e_G = e_F$, solid lines), setting $e_G < e_F$ (dashed lines) as proposed by Ringeisen
et al. (2021) has a very small impact on the monthly averaged drift speed.

Even though the non-normal flow rule does not remedy the problem of too large intersection angles, defining the flow rule
independently from the yield curve offers clear advantages. Studies have shown the impact of modifying $e = e_F = e_G$ (i.e., a
normal flow rule) on the simulation of landfast ice (Lemieux et al., 2016), ice arches (Dumont et al., 2009) and deformations
(Bouchat and Tremblay, 2017). But clearly, decreasing $e$ to enhance simulated landfast ice in coastal area also impacts the
deformations, the drift and the sea ice production everywhere. Our results demonstrate that $e_F$ and $e_G$ could be used to opti-
mize the representation of different physical processes. Modifying the yield curve by changing $e_F$ impacts simulated landfast
ice (and ice arches) and to a certain extent the drift of pack ice while modifying the flow rule by changing $e_G$ affects the
deformations (and their spatial scaling properties) and indirectly sea ice production through leads.

## 7 Concluding remarks

As demonstrated in other studies, we find that the elastic-viscous-plastic (EVP) rheology with a normal flow rule (i.e., $e_G = e_F$)
leads to a PDF of intersection angles (for conjugate pairs) with a peak at $90°$. Using a non-normal flow rule with the elliptical
yield curve, as introduced by Ringeisen et al. (2021), does not remedy this problem. Either with $e_G < e_F$ or $e_G > e_F$, the peak
is still at $90°$.





From VP experiments with a normal flow rule, Hutter and Losch (2020) compared the mean orientation of LKFs with the grid axis and could not find a clear correlation. For the SIREx project, Hutter et al. (2022) made the qualitative argument that the peak of the PDF at 90° is not caused by LKFs aligned with the grid as models with an unstructured grid also lead to a peak at 90°. Our analysis, involving the computation of the intersection angle for all the conjugate pairs, clearly indicates a tendency of simulated LKFs to be aligned with the grid.

But why Ringeisen et al. (2021), consistent with theory, were able to simulate smaller fracture angles (by setting $e_G < e_F$) in their idealized uniaxial compression experiments? We explain as follows this contradiction between our results with realistic pan-Arctic experiments and the ones of Ringeisen et al. (2021). We don't think it is related to the use of a C-grid and a VP rheology by Ringeisen et al. (2021) as opposed to the B-grid with EVP dynamics used here. Note that our conclusions remain the same whether the implicit approach (VP) in CICE or the explicit one (EVP) is used. We speculate that in the pan-Arctic simulations, a deformation (or fracture) that would be loosely aligned with a grid axis would tend to lock itself to that very same grid axis. This locking mechanism was not observed by Ringeisen et al. (2021) because their simulated fractures exhibit large angles with respect to the grid axis (see for example Fig. 6 in Ringeisen et al. (2021)). Understanding and proving the existence of this locking mechanism is beyond the scope of this paper and will be investigated as part of future work. To obtain simulated LKF intersection angles more in line with observations, we argue that not only the rheology needs to be improved but that more advanced numerical methods (i.e., spatial discretization, numerical solver, etc.) should be developed.

Although the non-normal flow rule does not remedy the problem of too wide simulated intersection angles, our results show that $e_G$ could be an additional tuning knob (independent of $e_F$) for adjusting statistics of LKFs and more generally deformations. A smaller $e_G$ increases the amount of divergence/convergence in plastic deformations; LKFs are also more numerous and better defined. As more leads are simulated with a smaller $e_G$, there is more sea ice formation which notably impacts the sea ice volume. We argue, however, that this impact would be mitigated in simulations using a coupled atmospheric model. On the other hand, the parameter $e_F$, which defines the shear strength associated with the yield curve, could be used to control the formation of ice arches and more generally the simulated landfast ice area and to a smaller extent to optimize the pack ice drift.

## Appendix A: Algorithm for calculating LKF width

Given a detected LKF with $N$ points (with indices $n = 0$ to $n = N - 1$), the algorithm for calculating the LKF width is applied for all these points. Coordinates $i, j$ are associated with all these LKF points. Fig. A1 illustrates parts of the procedure.

For a certain point (in orange) along the LKF, the algorithm calculates the local slope of the LKF with respect to the $i$ axis in order to define a unit vector aligned with the LKF (pointing in the direction of increasing $n$). The dark blue cells in Fig. A1 are the ones used to determine the slope and the unit vector aligned with the LKF. Away from the start and end points of the





LKF (panels a, b, c, e and f), two dark blues cells are used and the slope is obtained from a central finite difference. However,
at the start point, the orange and dark blue cell are used to calculate a one-sided first-order approximation of the slope in order
to obtain the unit vector aligned with the LKF (panel d). Similar approaches are used for the end point and for diagonal LKFs
(not shown).

Two other unit vectors are then defined at 90° from the unit vector aligned with the LKF. These two vectors define the search
direction for calculating the LKF width. Note that some points of a detected LKF can lead to ambiguous search directions. An
example of this is in panel f of Fig. A1. For these points, the width is not calculated and is not used to calculate metrics such
as the mean width of LKFs.

The thinning procedure of the LKF detection algorithm identifies the LKF as a series of points each corresponding to the
largest total deformation along a perpendicular transect. Given the location of a LKF point with a total deformation of $\dot\epsilon_{tot}$, the
width calculation algorithm calculates the number of grid cells (or pixels) required in both perpendicular search directions for
reducing the total deformation below $\alpha\dot\epsilon_{tot}$ where $0 < \alpha < 1$. The parameter $\alpha$ is set to 0.5 in this study.

The width of the LKF at a certain point is then the sum of the two half-widths associated with the two search directions.
Consider a point $i, j$ of a LKF along the $x$ axis (with search directions along the $y$ axis). If $\dot\epsilon_{tot}(i, j + 1) > \alpha\dot\epsilon_{tot}(i, j)$ and
$\dot\epsilon_{tot}(i, j + 2) < \alpha\dot\epsilon_{tot}(i, j)$, the half-width is 2 pixels along this search direction. Similarly, for a point $i, j$ of an oblique LKF,
$\dot\epsilon_{tot}$ values at $i + 1, j + 1$, $i + 2, j + 2$, ... are compared to $\dot\epsilon_{tot}(i, j)$. If $\dot\epsilon_{tot}(i + 2, j + 2)$ is the first value below $\alpha\dot\epsilon_{tot}(i, j)$, the
half-width is $2\sqrt{2}$ pixels. Note that for each search direction, $\dot\epsilon_{tot}$ is compared to $\alpha\dot\epsilon_{tot}(i, j)$ up to a maximum number of grids
cells defined by $c_{max}$. Hence, as an example for the horizontal LKF, if $\dot\epsilon_{tot}(i, j + c_{max}) > \alpha\dot\epsilon_{tot}(i, j)$, the procedure is stopped
and the half-width is set to $c_{max}$ pixels. The parameter $c_{max}$ is set to 5 is this study.

## Appendix B: Algorithm for calculating intersection angles and angles with the grid

The algorithm for calculating intersection angles of conjugate pairs first identifies intersecting LKFs. Consider the example in
Fig. B1 which shows two intersecting LKFs: LKF1 in blue and LKF2 in yellow. LKF1 has $N_1$ detected points (with indices
$n = 0$ to $n = N_1 - 1$) and LKF2 has $N_2$ points (with indices $n = 0$ to $n = N_2 - 1$). The intersection point (the orange cell)
has index $n = n_{i1}$ for LKF1 and $n = n_{i2}$ for LKF2. To determine if LKF1 and LKF2 represent a conjugate pair, the algorithm
defines local regions for both LKFs for calculating the mean vorticity and for fitting a polynomial of degree 1 (i.e., a line). The
region for the polynomial fits are defined by $n_{min1}, n_{max1}$ for LKF1 (same idea for LKF2) with $n_{min1} = \max(0, n_{i1} - \Delta_n)$
and $n_{max1} = \min(n_{i1} + \Delta_n, N_1 - 1)$. This local region is represented by the dark blue cells for LKF1 and by the dark yellow
cells for LKF2. While $\Delta_n$ is set to 10 for the results of this article, it is for simplicity assumed to be equal to 4 in the example
displayed on Fig. B1. Because the top-right cell of LKF2 is the last detected point of this LKF, there are only three LKF points



on the 'right side' of the orange cell for the polynomial fit of LKF2. The polynomial fits are represented by the dashed gray lines.

The vorticity is then analyzed for LKF1 in the region of intersection (defined by $n_{min1}, n_{max1}$) and for LKF2 for $n_{min2}$ to $n_{max2}$. If LKF1 and LKF2 have mean vorticities of opposite sign, they are interpreted as a conjugate pair of LKFs. The angle $\theta_c$ is then calculated counterclockwise from the fitted LKF with positive vorticity to the fitted LKF with negative vorticity. In our example, LKF1 (in blue) has a positive mean vorticity while LKF2 (in yellow) has a negative mean vorticity. The angle $\theta_c$ is acute in this case.

Given an identified conjugate pair as described above, the second part of the algorithm calculates the angles of the conjugate LKFs with a $x - y$ coordinate system defined at the intersection point. The $x$ and $y$ axis of this coordinate system are locally aligned with the computational grid (at the tracer point). Fig. B2 explains how these angles are calculated. The LKF (yellow cells) is LKF2 in the example above for the conjugate angle. Again, the orange cell represents the intersection point between the conjugate pair LKF1 and LKF2. The dashed gray line is the polynomial fit for LKF2 in the local region of the intersection
point. Given the slope $m_2$ of the polynomial fit of LKF2, the acute angle $\theta_x$ is computed as

$$\theta_x = \tan^{-1}\left|\frac{m_2 - m}{1 + m_2 m}\right|, \tag{B1}$$

where $m = 0$ is the slope of a line on the $x$ axis.

A similar calculation is done for $\theta_y$. We finally calculate $\theta_{min}$, the minimum angle, which is simply equal to $\min(\theta_x, \theta_y)$.
The same process is then repeated for the other conjugate LKF (i.e., LKF1).

*Code and data availability.* The simulations for this article were done with release 6.5.0 of the CICE sea ice model which can be obtained at https://github.com/CICE-Consortium/CICE/releases/tag/CICE6.5.0 and on Zenodo at https://doi.org/10.5281/zenodo.10056499. Release 6.5.0 includes Icepack 1.4.0. The LKF detection and analysis package can be downloaded from https://github.com/JFLemieux73/lkf_tools.
The PIOMAS sea ice volume dataset can be obtained at https://psc.apl.uw.edu/research/projects/arctic-sea-ice-volume-anomaly/data.

*Author contributions.*

JFL implemented the plastic potential approach in CICE. FR, PB and JFL setup the ice-ocean model configuration. JFL, MP, BT, NH and DR designed the numerical experiments. JFL ran the numerical experiments. NH provided guidance on the use of the LKF detection algorithm. JFL developed the LKF analysis package with contributions from NH and MP. MP developed





the code and produced the figure for the spatial scaling analysis. JFL wrote the article with contributions from all the co-authors.

*Competing interests.*  no competing interests are present





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





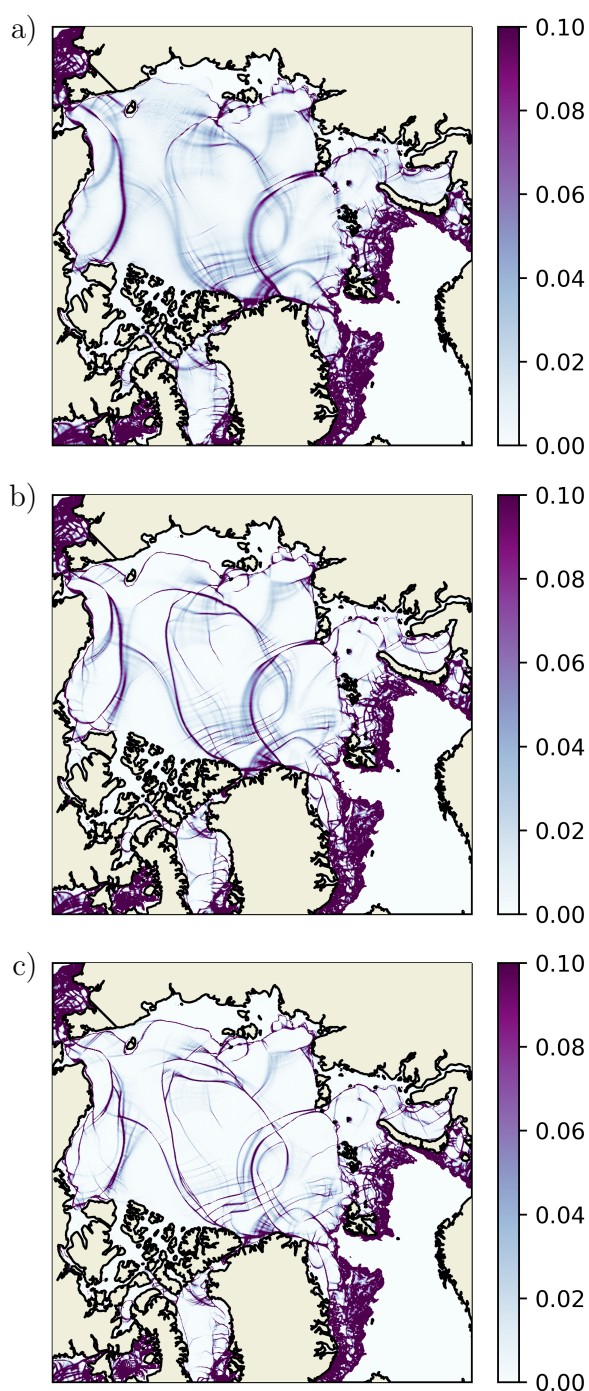

**Figure 1.** Total deformation in day$^{-1}$ on 25 April 2005 for $e_\mathrm{F} = 1.75$ and $e_\mathrm{G} = 2.63$ (a), $e_\mathrm{G} = 1.75$ (b) and $e_\mathrm{G} = 1.16$ (c). Deformations are not displayed for a concentration smaller than 0.15.





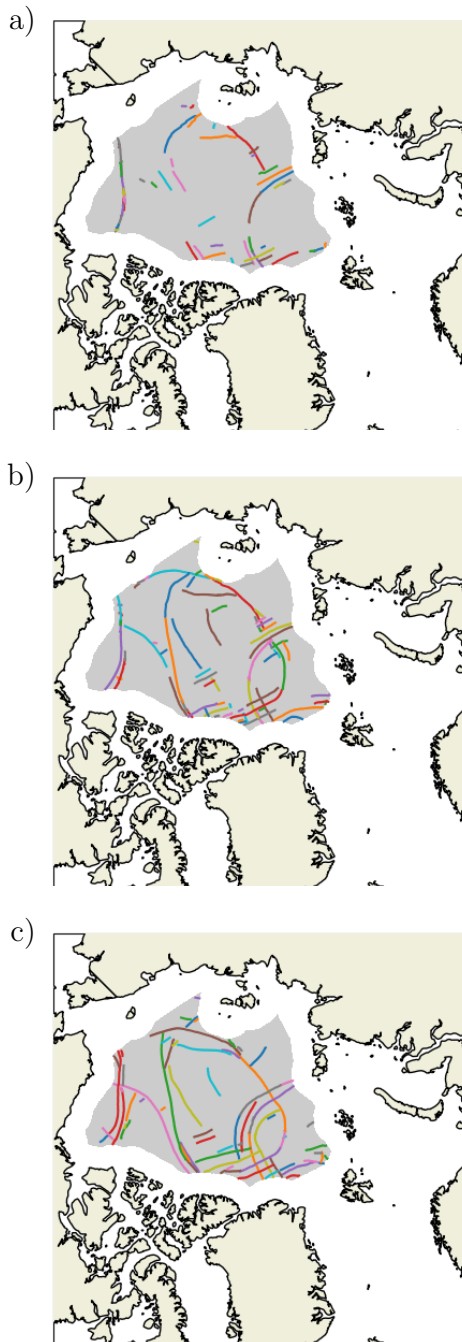

**Figure 2.** Detected LKFs on 25 April 2005 for $e_F = 1.75$ and $e_G = 2.63$ (a), $e_G = 1.75$ (b) and $e_G = 1.16$ (c). The Arctic mask defining the region for the LKF detection is clearly visible in these panels.




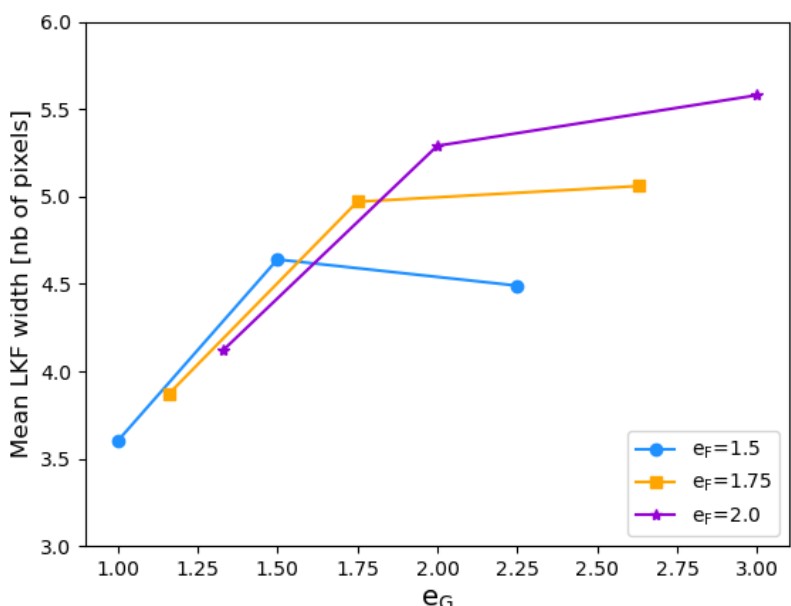

**Figure 3.** Mean width of detected LKFs for the period 1 January to 31 May 2005.

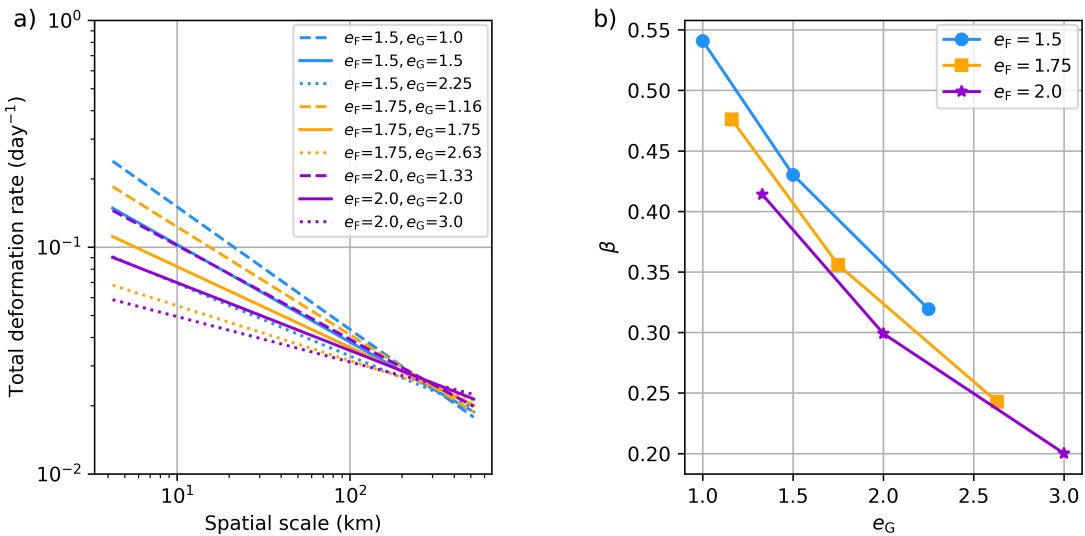

**Figure 4.** Spatial scaling of the total deformation rates for each of the main experiments (a) and corresponding scaling exponents as function of $e_G$ (b), for the period 1 January to 31 May 2005.





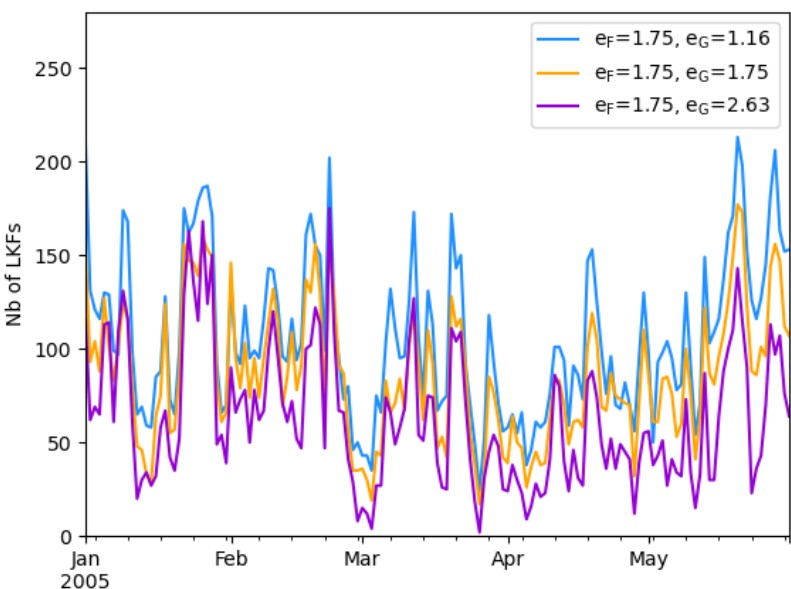

**Figure 5.** Number of detected LKFs as a function of time for different values of $e_F$ and $e_G$.

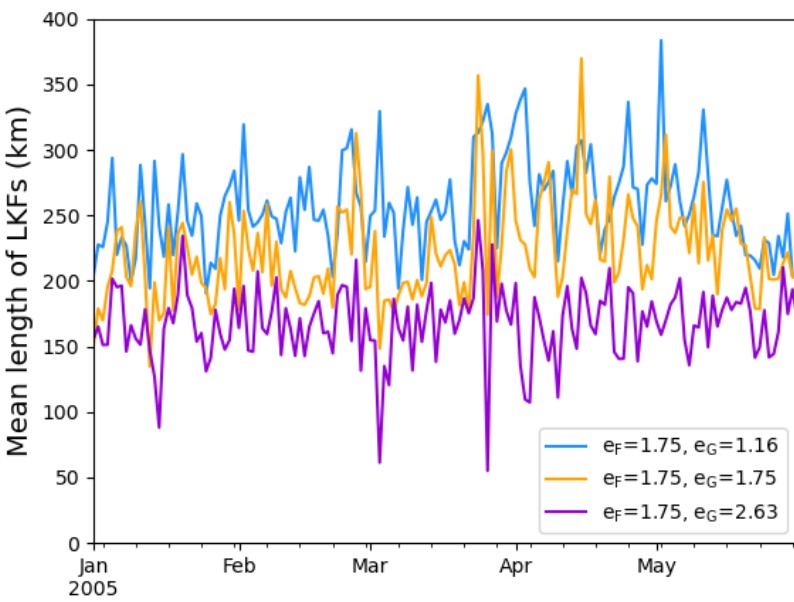

**Figure 6.** Mean length in km of detected LKFs as a function of time for the period 1 January to 31 may 2005.





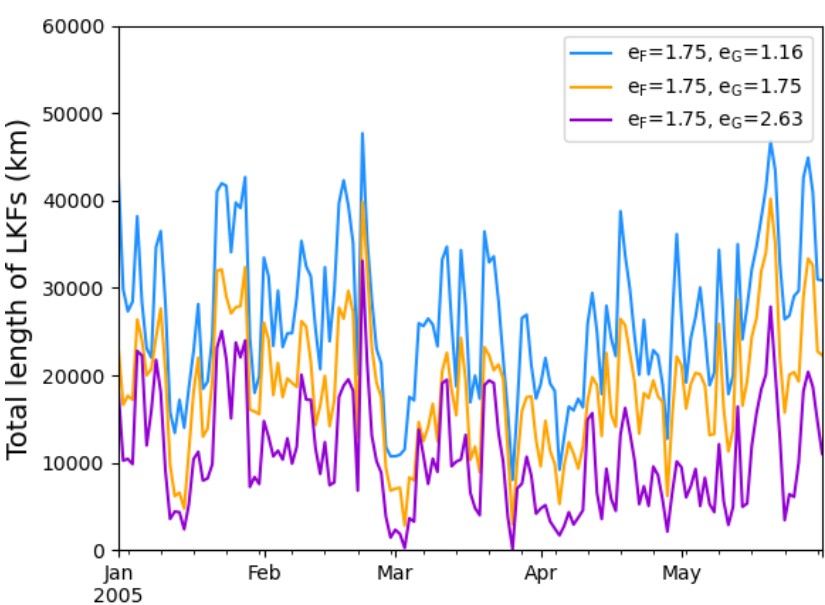

**Figure 7.** Total length in km of detected LKFs as a function of time for the period 1 January to 31 may 2005.



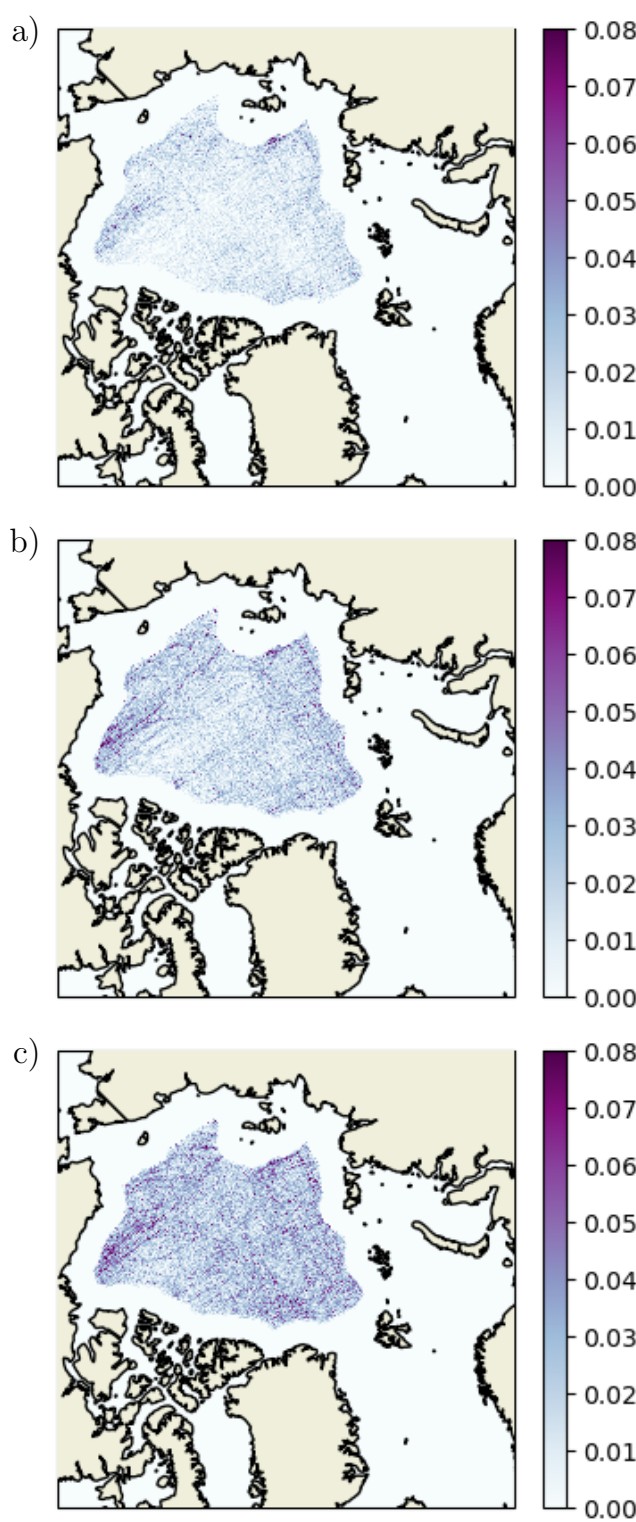

**Figure 8.** LKF density for the period 1 January to 31 may 2005 for $e_F = 1.75$ and $e_G = 2.63$ (a), $e_G = 1.75$ (b) and $e_G = 1.16$ (c).



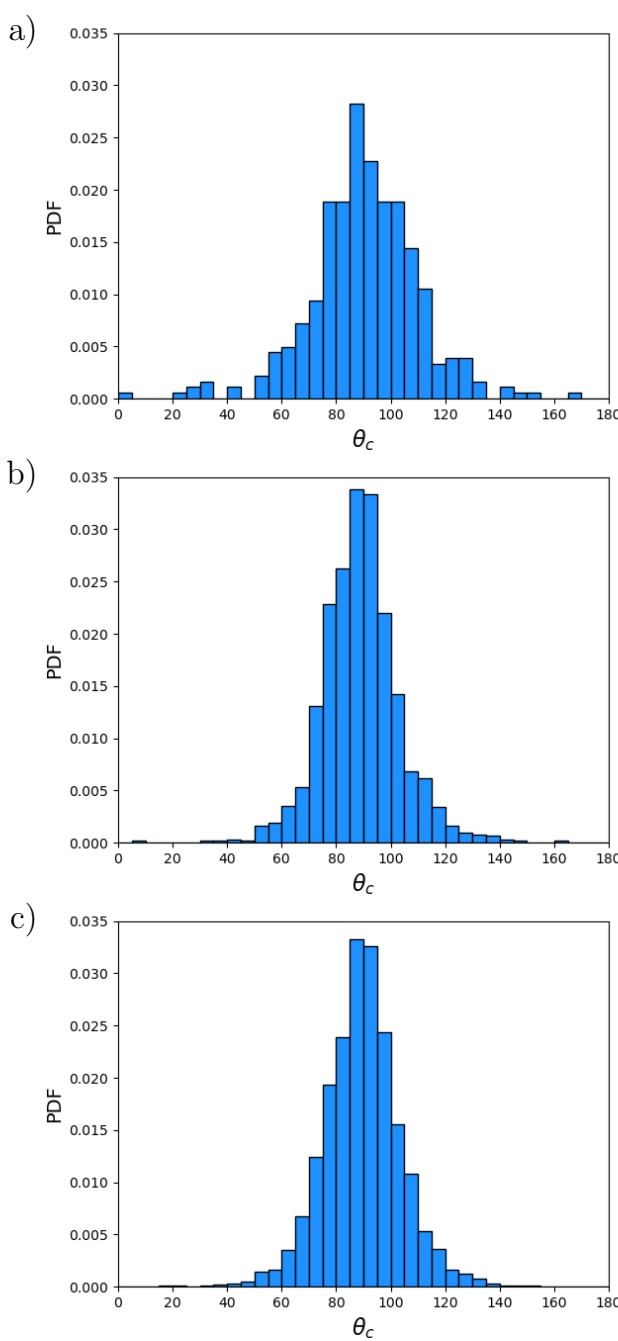

**Figure 9.** PDF of $\theta_c$, the intersection angle of conjugate pairs, for $e_F = 1.75$ and $e_G = 2.63$ (a), $e_G = 1.75$ (b) and $e_G = 1.16$ (c). The PDF is calculated from daily snapshots at 00 UTC for the period 1 January to 31 May 2005.



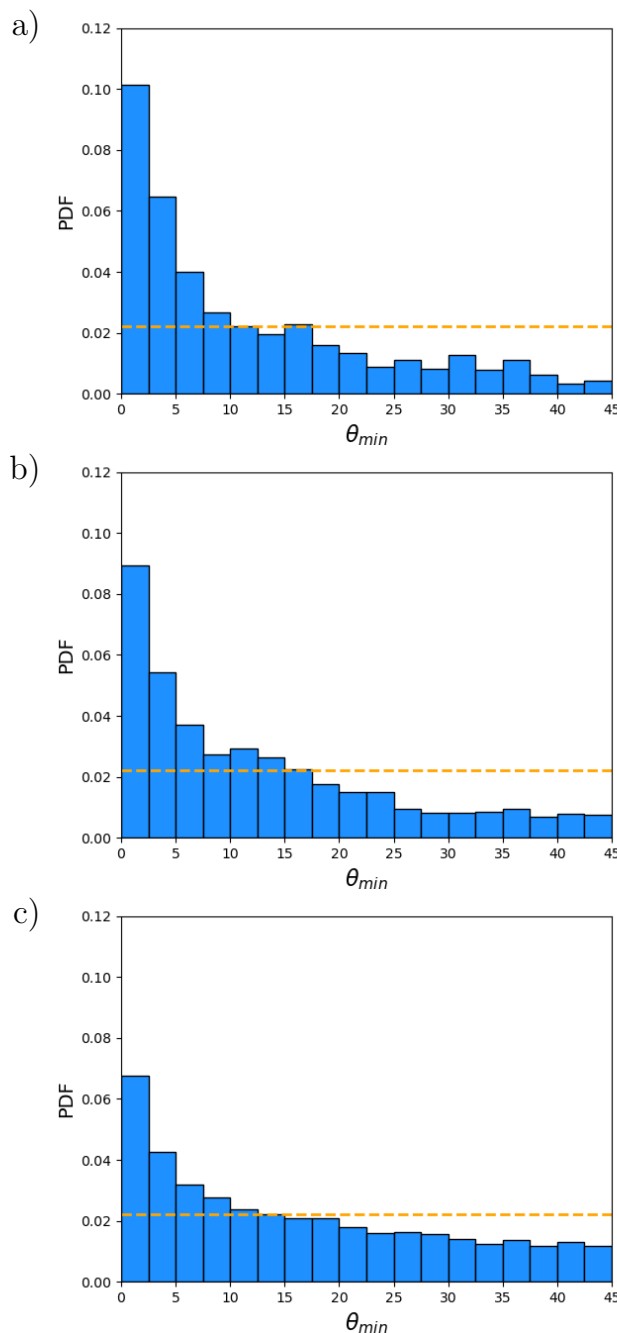

**Figure 10.** PDF of $\theta_{min}$, the minimum angle between detected LKFs of conjugate pairs and the $x$ and $y$ axis for $e_F = 1.75$ and $e_G = 2.63$ (a), $e_G = 1.75$ (b) and $e_G = 1.16$ (c). The angles are calculated at the intersection point of a conjugate pairs. The PDF is calculated from daily snapshots at 00 UTC for the period 1 January to 31 May 2005. The orange dashed line shows what would be the PDF if LKFs did not have a preferred orientation with respect to the grid.




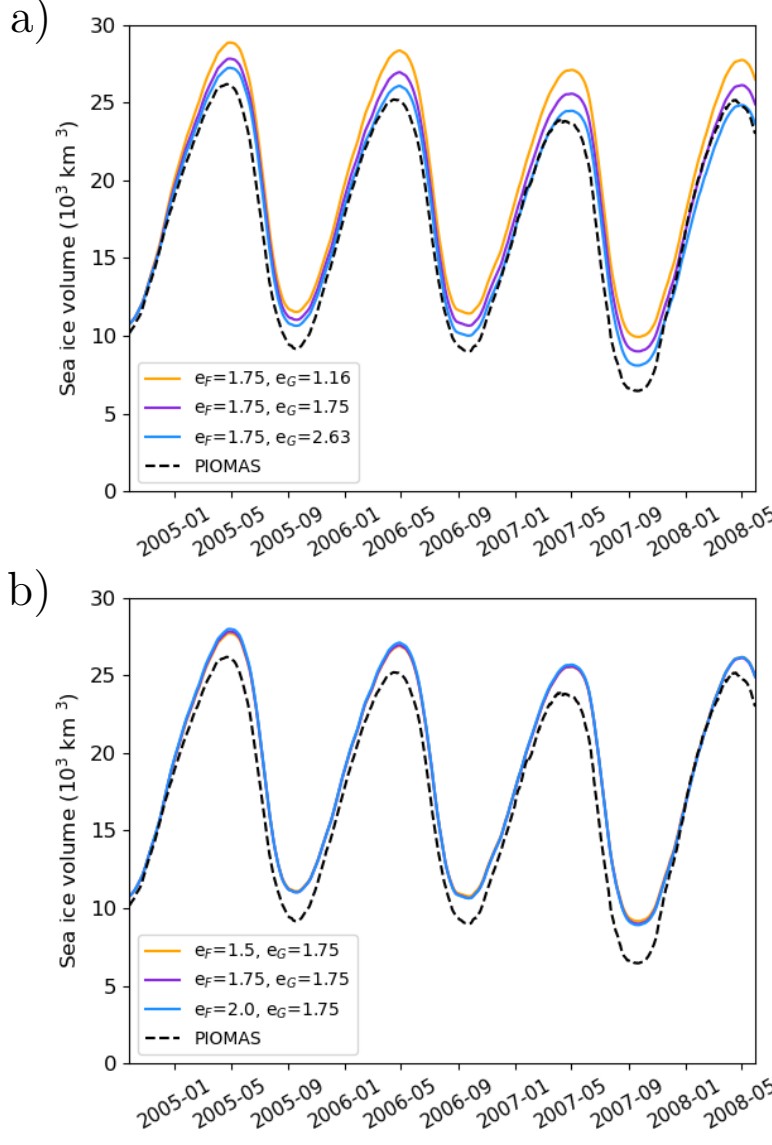

**Figure 11.** Total simulated daily mean sea ice volume as a function of time for different values of $e_F$ and $e_G$. As a comparison, the dashed black line shows the sea ice volume simulated by the PIOMAS system.



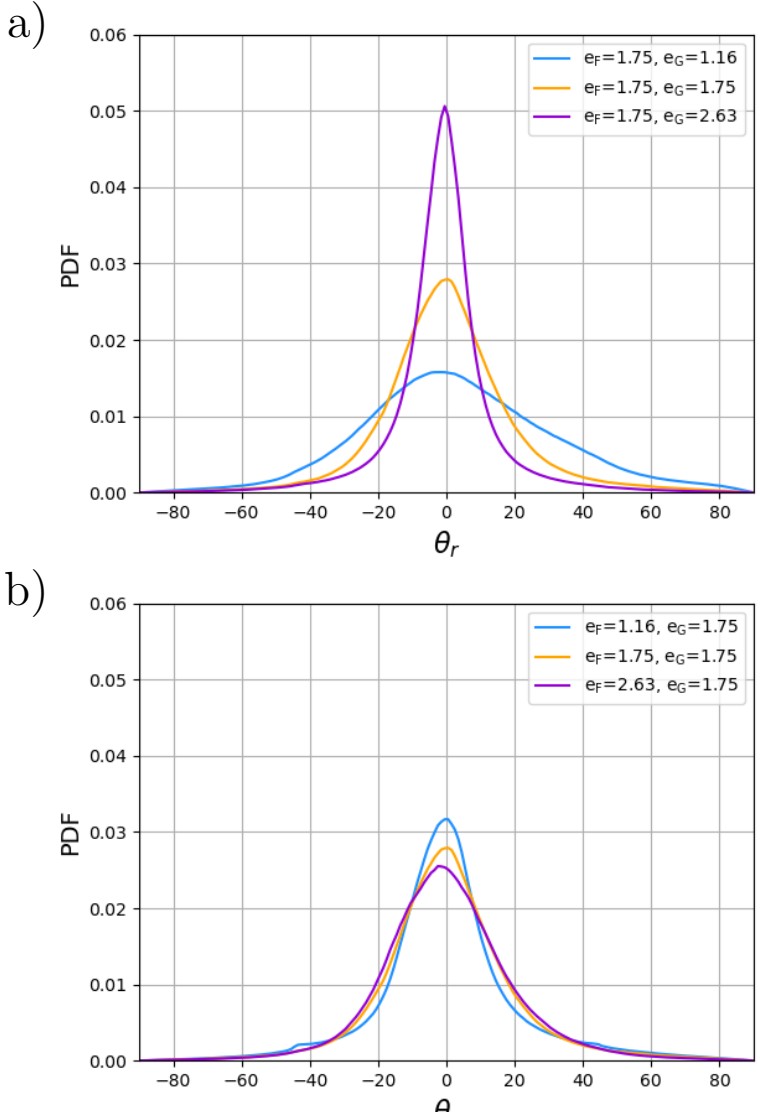

**Figure 12.** a) PDF of $\theta_r$ for a fixed value of $e_F = 1.75$ and $e_G = 1.16$ (blue), $e_G = 1.75$ (orange), and $e_G = 2.63$ (violet) and $e_F = 1.5$ (blue). b) PDF of $\theta_r$ for a fixed value of $e_G = 1.75$ and $e_F = 1.5$ (blue), $e_F = 1.75$ (orange), and $e_F = 2.0$ (violet). $\theta_r$ characterizes the ratio of div over shear (need to be defined). The PDFs are calculated from daily snapshots at 00 UTC for the period 1 January to 31 May 2005.



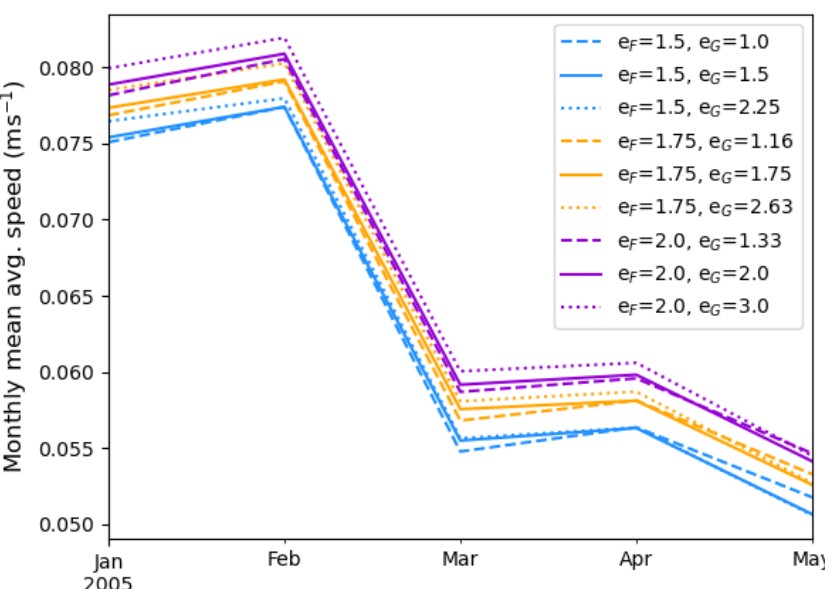

**Figure 13.** Monthly mean of daily averaged sea ice speed for the period 1 January to 31 May 2005 calculated over the pack ice mask region. The daily averaged sea ice speed is calculated where the daily averaged concentration is larger than 0.15.





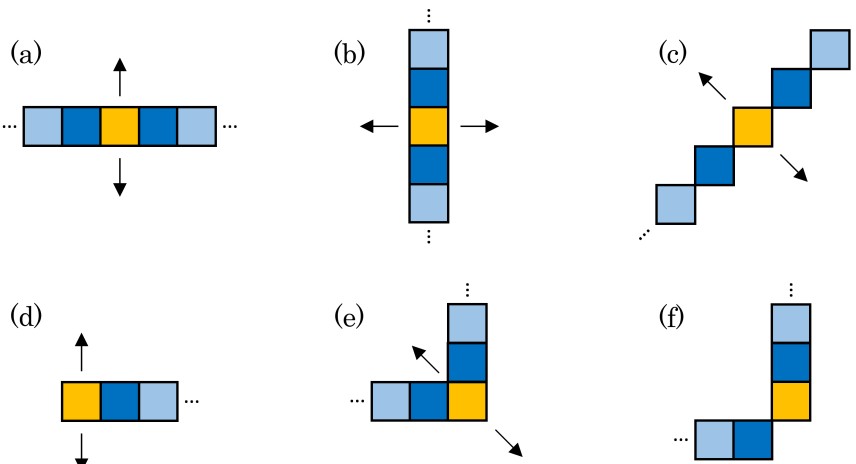

**Figure A1.** Schematic explaining the search direction algorithm for estimating LKF width for different LKF shapes and orientations. For a given case, the width is calculated from the orange cell. The dark blue cells are used to determine the local direction of the LKF; Light blue cells are not involved. Given a vector locally aligned with the LKF, the search directions (black arrows) are defined at 90° from this vector. There is a case similar to (c) (i.e. at 90°) that is not shown. There are also similar cases to (d) and (e) that are not displayed. The width is not calculated for case (f) (same idea for similar cases) because the search directions are ambiguous.



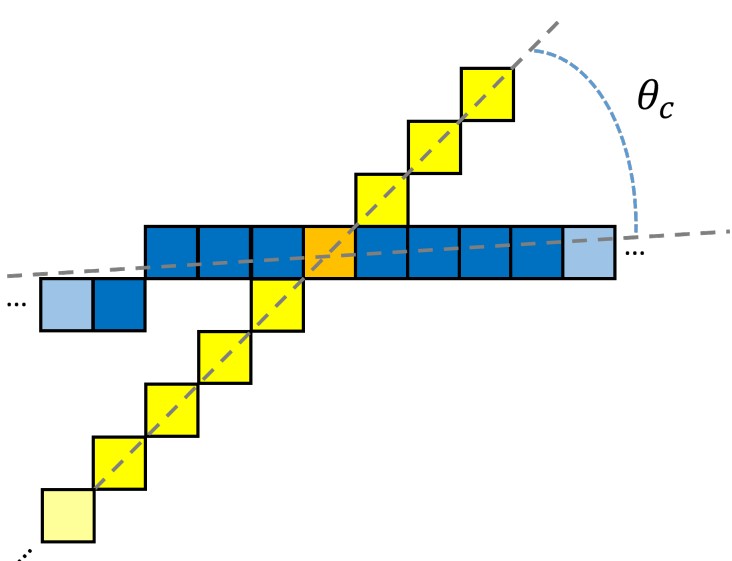

**Figure B1.** Schematic explaining the algorithm for measuring intersection angles for conjugate pairs. The orange cell represents the intersection point between LKF1 (in blue) and LKF2 (in yellow). The dark blues cells are used for the local polynomial fit for LKF1 while the dark yellow ones are used for LKF2. Light blue and light yellow cells are part of the detected LKFs but are not used for the polynomial fits. $\Delta_n = 4$ points are used for LKF1 on each side of the intersection points. However, only 3 points are used for LKF2 on one side as the top-right cell is the last point of this LKF. The dashed gray lines are the two polynomial fits. In this example, it is assumed that LKF1 has a positive mean vorticity while LKF2 has a negative mean vorticity (in the local region). The angle $\theta_c$ is measured from the polynomial fit for LKF1 to the polynomial fit for LKF2; the angle is acute in this example.




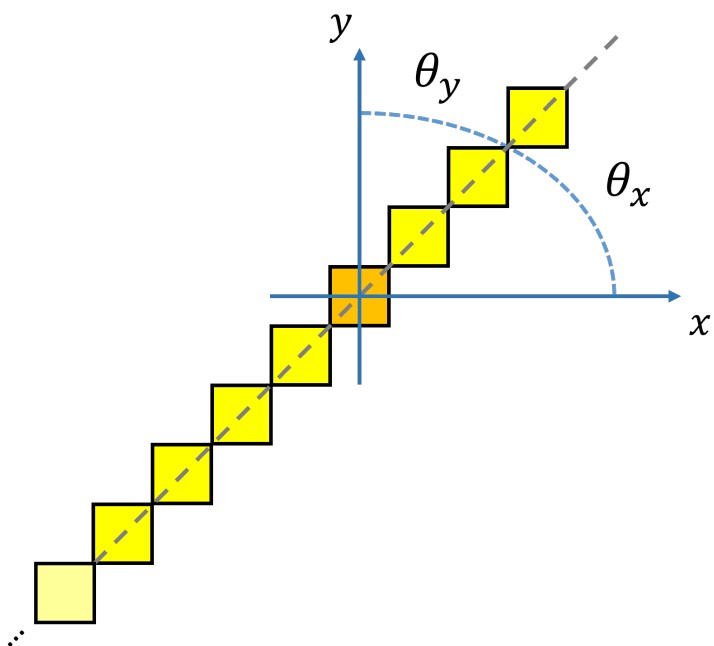

**Figure B2.** Schematic explaining the algorithm for measuring the angle of conjugate pair LKFs with the $x$ axis ($\theta_x$) and with the $y$ axis ($\theta_x$). The orange cell represents the intersection point between the conjugate pair LKF1 and LKF2. For clarity, only LKF2 is shown (in yellow). The dashed gray line is the polynomial fit for LKF2 in the local region of the intersection point. The acute angles $\theta_x$ and $\theta_y$ are measured with respect to a local coordinates system with its origin at the intersection point of the conjugate pair. The same process is repeated for the other conjugate LKF (i.e., LKF1).