# Peer review of "Impact of non-normal flow rule on linear kinematic features in pan-Arctic ice-ocean simulations"

_EGUsphere, 2024_

## Referee Comment (RC1)

**Review of the Paper: "Impact of Non-Normal Flow Rule on Linear Kinematic Features in Pan-Arctic Ice-Ocean Simulations"**
*Jean-François Lemieux, Mathieu Plante, Nils Hutter, Damien Ringeisen, Bruno Tremblay, François Roy, and Philippe Blain*
* * *
**Summary:**
This paper presents a detailed sensitivity study on the influence of a non-normal flow rule in the viscous-plastic (VP) sea-ice rheology on the simulation of linear kinematic features (LKFs) in Arctic sea ice. The authors implement the concept of a plastic potential and non-normal flow rule introduced by Ringeisen et al. (2021) into the CICE sea ice model and conduct a comprehensive series of pan-Arctic simulations based on this implementation.

However, the central result of the study is that the peak of the probability density function (PDF) of the simulated intersection angles between conjugate LKFs remains at 90°, whereas observational data show a peak around 45°. This finding is counterintuitive, as Ringeisen et al. (2021) demonstrated that using a non-normal flow rule enables the simulation of smaller intersection angles that align more closely with observations. This discrepancy raises a fundamental and scientifically important question: what is the underlying cause of this difference? Answering this question would be interesting and valuable for the community.

Instead of systematically addressing this question, the authors limit themselves to a purely parametric sensitivity analysis. As a result, parts of the paper take on the character of a technical documentation of the non-normal flow rule implementation in CICE, without achieving the intended effect, namely more realistic intersection angles. In my view, the innovative contribution of this paper is therefore limited.

Since the discrepancy between the idealized tests in Ringeisen et al. (2021) and the CICE implementation remains unresolved, the overall validity of the presented results is questionable. In my view, it is entirely possible that the peak in LKF intersection angles at 90 degrees arises from numerical details specific to the CICE implementation, and that a careful reproduction of the setup proposed by Ringeisen et al. could yield a peak at smaller angles.

The authors' explanation, that this discrepancy is due to a grid effect, appears vague and insufficiently analyzed. Moreover, this hypothesis is not in line with previous studies, such as Hutter et al. (2022) and Hutter and Losch (2020).

**Recommendation:**

I therefore recommend that the manuscript place greater emphasis on the core scientific question: Why does the CICE implementation yield different results from the idealized tests by Ringeisen et al.? A targeted and transparent validation of the implementation is essential in this regard. For these reasons, I recommend a **major revision**.
* * *
**Main Points:**

1. The authors write:
   "We don't think it is related to the use of a C-grid and a VP rheology by Ringeisen et

al. (2021) as opposed to the B-grid with EVP dynamics used here."

To validate this claim and, more importantly, to demonstrate that the implementation performs as intended, I suggest reproducing the idealized test case from Ringeisen et al. within CICE and first evaluating whether their results can be replicated. This would confirm that the basic implementation is correct and that differences in numerical treatment—such as C-grid vs. B-grid or the choice of solver—do not, as the authors suggest, contribute to the observed deviations in the results.

2. "For VP experiments with a normal flow rule, Hutter and Losch (2020) compared the mean orientation of LKFs with the grid axis and could not find a clear correlation. For the SIREx project, Hutter et al. (2022) made the qualitative argument that the peak of the PDF at 90° is not caused by LKFs aligned with the grid as models with an unstructured grid also lead to a peak at 90°. Our analysis, involving the computation of the intersection angle for all the conjugate pairs, clearly indicates a tendency of simulated LKFs to be aligned with the grid."

This is already the second aspect in which the results of this study seem to differ from existing work. To relate the observed behavior to a grid effect, a more in-depth investigation, as suggested in point 1, is required.

3. In line 351, the authors write:

"As demonstrated in other studies, we find that the elastic-viscous-plastic (EVP) rheology with a normal flow rule (i.e., eG = eF) leads to a PDF of intersection angles (for conjugate pairs) with a peak at 90°. Using a non-normal flow rule with the elliptical yield curve, as introduced by Ringeisen et al. (2021), does not remedy this problem. Either with eG < eF or eG > eF, the peak is still at 90°."

In my opinion, this study cannot claim general validity of its findings, as it remains unclear what the cause of the observed failure is. In particular, it has not been convincingly ruled out that the observed effect may be due to details in the implementation. From my perspective, it is still possible that the effect observed by Ringeisen can also be reproduced in pan-Arctic simulations, provided that the setup is carefully implemented.
* * *
**Minor Points:**

**Introduction:**

- **Line 32**: "It is expressed with the use of a plastic potential: the post-failure deformations are normal to the plastic potential."
  This sentence is extremely technical for an introduction and difficult to understand. Some readers may not know what it means that post-failure deformations are normal to the plastic potential. It should either be reworded and better explained or supplemented with a sketch or equation that clarifies the meaning.
- **Lines 31–40**: These are too technical and difficult to digest for an introduction. Please revise and invest more effort in explaining the technical terminology.

**Section 2:**

- Move Section 2 to the appendix. The presentation of the concepts is currently very isolated, making it completely unclear how they relate to the equations that describe the rheology.

- For example, in **line 60**: Which system of equations are you referring to? Please specify clearly.
- In addition to the equations describing the rheology or yield curve in the standard VP model and in Ringeisen's version, graphical representations would be helpful for better understanding the key parameters **eF, eG, and e**. In the current presentation, it is very difficult for readers from the Cryosphere community, who may not be familiar with all of Ringeisen's previous work, to follow the argument.
    - **Line 65**: Please explicitly state the constitutive equation.
    - **Line 77**: "With the non-normal flow rule, η is now equal to $e^{-2}\zeta$." – Why? Please explain.

**Methodology:**

- **Line 136**: Where does this choice come from? Why is eG set to ~ eF/1.5, eF, or ~ 1.5eF? Please justify.
- **Line 155**: "A kernel value of seven is used for the detection algorithm." – Why 7? What does it mean? Please explain.
- **Lines 168–173**: What is α? What does "morphological thinning" mean? Please include a sketch to make this clearer.
- **Line 176**: What do you mean by "conjugate fault lines"?
- **Line 177**: What exactly is the vorticity being analyzed?
- **Lines 177–182**: This section is extremely technical and hard to understand. Either explanatory sketches should be added, or the content should be moved to the appendix.

**Figure 1a:**

- To me, the structures shown look like numerical instabilities. It is possible that the modification proposed by Ringeisen increases the stiffness of the system of equations, making it harder to solve. While 900 iterations are considered sufficient in the standard EVP model, they may be inadequate for the modification investigated here. I therefore recommend conducting tests with smaller time steps and/or more iterations to check whether sharper structures can be resolved and whether the 900 EVP subcycles are indeed sufficient here.

**Results:**

- Line 366: "Note that our conclusions remain the same whether the implicit approach (VP) in CICE or the explicit one (EVP) is used."
  There are numerous publications, including some by Lemieux, that show that ten Picard steps are insufficient to achieve a convergent solution. Therefore, it is possible that the observed effect is due to the inadequate accuracy of the approximation solutions in both setups (EVP and Picard).

---

## Referee Comment (RC2)

This is a review of the manuscript entitled *"Impact of non-normal flow rule on linear kinematic features in pan-Arctic ice-ocean simulations."* The study discusses the effect of allowing the rheology used in the CICE large-scale sea ice model to deviate from the normal flow rule, which is normally used to determine the direction of deformations when sea ice undergoes plastic deformation. It is largely motivated by a previous study that showed that using a non-normal flow rule resulted in more realistic linear kinematic features (LKFs) in a more idealized setup. In particular, it had been shown to significantly improve the model's ability to reproduce realistic intersection angles of conjugate faults. Here, the authors show that this is not the case. They discuss the potential reasons for this negative result compared to the more idealized study and suggest (with some supporting arguments) that these intersection angles may be largely constrained to follow the model grid axes, resulting in intersection angles often close to 90° in the case of the grid used here. Despite this negative result, they show that using a non-normal flow rule has other effects on LKF properties, which could make its use a convenient way to tune LKF characteristics in large-scale sea ice models.

The manuscript is well-written and clear. The scope is well-defined, and the analysis is sound and well-supported. The topic is perhaps a bit niche, but it aligns with previous studies published in the journal and is of interest to the sea ice modelling community targeted by *The Cryosphere*. The suggestion that the peak at 90deg partly results from the alignments of LKFs with the grid axes is sound. It may not be fully demonstrated here (as mentionned), but the authors have gathered enough elements to point to this cause in their results and discussion. The argument is strong enough to motivate the community interested in sea ice deformations to investigate the numerics instead of only focusing on the physics. Additionally, the fact that the non-normal flow rule can be "tweaked" to tune sea ice deformations properties is an interesting result for the sea ice modelling community. Therefore, I support the publication of this manuscript in *The Cryosphere*. I do not have comments that I consider major, but I would recommend the minor revisions below:

**General comments:**

Given that one of the main results is that using a non-normal flow rule can be a convenient way to tune certain LKF properties independently, the authors may consider commenting on the implementation in CICE. Does it require substantial modification to the code? Does it significantly impact computational time?

I sometimes found the order in which the figures are referenced a bit confusing. For example, it appears that Figure 4 is referred to after Figures 5 and 7, and Figure 2 before Figure 1. This has little impact in the preprint, since all figures are at the end, but the authors may consider reordering the figures for the final version.

Specific comments:

L213: I think "of" is missing (or something else)
L337/338: I find this sentence a bit unclear.
L340/350: I find the ideas in this paragraph a bit hard to follow, I think it would benefit from a bit of rewriting (especially the sentence starting with "But clearly..")

Best regards,
Guillaume Boutin

---

## Author Comment (AC1)

**Initial response to reviewer 1 after first review**

26 May 2025

We thank the reviewer for his/her time and for his/her useful comments and suggestions. Our responses below are in blue.

**Reviewer 1**:

This paper presents a detailed sensitivity study on the influence of a non-normal flow rule in the viscous-plastic (VP) sea-ice rheology on the simulation of linear kinematic features (LKFs) in Arctic sea ice. The authors implement the concept of a plastic potential and non-normal flow rule introduced by Ringeisen et al. (2021) into the CICE sea ice model and conduct a comprehensive series of pan-Arctic simulations based on this implementation. However, the central result of the study is that the peak of the probability density function (PDF) of the simulated intersection angles between conjugate LKFs remains at 90°, whereas observational data show a peak around 45°. This finding is counterintuitive, as Ringeisen et al. (2021) demonstrated that using a non-normal flow rule enables the simulation of smaller intersection angles that align more closely with observations. This discrepancy raises a fundamental and scientifically important question: what is the underlying cause of this difference? Answering this question would be interesting and valuable for the community. Instead of systematically addressing this question, the authors limit themselves to a purely parametric sensitivity analysis. As a result, parts of the paper take on the character of a technical documentation of the non-normal flow rule implementation in CICE, without achieving the intended effect, namely more realistic intersection angles. In my view, the innovative contribution of this paper is therefore limited. Since the discrepancy between the idealized tests in Ringeisen et al. (2021) and the CICE implementation remains unresolved, the overall validity of the presented results is questionable. In my view, it is entirely possible that the peak in LKF intersection angles at 90 degrees arises from numerical details specific to the CICE implementation, and that a careful reproduction of the setup proposed by Ringeisen et al. could yield a peak at smaller angles. The authors' explanation, that this discrepancy is due to a grid effect, appears vague and insufficiently analyzed. Moreover, this hypothesis is not in line with previous studies, such as Hutter et al. (2022) and Hutter and Losch (2020).

We think our manuscript focused too much on the intersection angles and

did not convey enough that, beside the intersection angles, a non-normal flow rule is a very interesting and useful model capability. Indeed, the plastic potential allows to optimize the deformations while maintaining the same yield curve. Similarly, the yield curve can be modified to optimize simulated landfast ice and sea ice drift with little impact on the deformations. We will rephrase some of the text to emphasize these clear advantages of the plastic potential.

I therefore recommend that the manuscript place greater emphasis on the core scientific question: Why does the CICE implementation yield different results from the idealized tests by Ringeisen et al.? A targeted and transparent validation of the implementation is essential in this regard. For these reasons, I recommend a major revision.

As described below, we have verified that our implementation of the plastic potential is consistent with the one of Ringeisen et al. 2021. Giving a complete explanation why the non-normal flow rule with $e_G < e_F$ does not lead to more realistic LKF intersection angles in pan-Arctic experiments is beyond the scope of this paper. In the mean time, we think that our study is sound and could inspire other researchers to further investigate this.

Major comments:

1) The authors write: "We don't think it is related to the use of a C-grid and a VP rheology by Ringeisen et al. (2021) as opposed to the B-grid with EVP dynamics used here." To validate this claim and, more importantly, to demonstrate that the implementation performs as intended, I suggest reproducing the idealized test case from Ringeisen et al. within CICE and first evaluating whether their results can be replicated. This would confirm that the basic implementation is correct and that differences in numerical treatment—such as C-grid vs. B-grid or the choice of solver—do not, as the authors suggest, contribute to the observed deviations in the results.

As stated in our response to reviewer 2, implementing the plastic potential of Ringeisen et al. 2021 is incredibly simple; it requires minor modifications to the code. We have revised our code and confirm it is consistent with the plastic potential formulation of Ringeisen et al. 2021.

We have also briefly investigated whether we can reproduce the results of Ringeisen et al. (2021) in idealized experiments. We have developed a test similar to the one described in Ringeisen et al. (2021). In our test, the ice is compressed against the eastern wall. Implementing the Dirichlet boundary conditions in CICE for the velocity on the western edge is quite complicated. To ease the implementation of this test in CICE and still mimic the experiments of Ringeisen et al. (2021), we used at the western edge a narrow block (5 grid cells) of very thick ice (5 m) forced by very strong winds (48 m/s). Fig. 1 shows the simulated shear deformation after one hour of simulation for $e_F = 2$

and $e_G = 1.4$. This is similar to Fig. 6a in Ringeisen et al. (2021). Damien Ringeisen, co-author here, previously used similar setups to successfully simulate fractures in uniaxial compression experiments. The fracture lines on the left look slightly bended compared to the ones obtained in Ringeisen et al (2021). This is expected given the different timesteps, length of simulation, numerical convergence, as well as differences in forcing timing, type and magnitude. See for example Fig 6 in Ringeisen et al. (2019). Nevertheless, we use the fracture lines on the right to calculate the intersection angles.

We therefore computed the fracture angles for $e_F = 2$ and $e_F = 4$ for different values of $e_G$. Fig. 2 below summarizes the simulated angles for all our numerical experiments and shows the Arthur angles, the theoretical angles that fit the best the results of Ringeisen et. (2021). This figure can be compared to Fig. 7a in Ringeisen et al. (2021). For $e_F = 2$, the simulated fracture angles are remarkably similar to the ones obtained by Ringeisen et al. (2021) and are close to the theoretical angles. Although our fracture angles for $e_F = 4$ are slightly larger than the Arthur angles and the ones of Ringeisen et al. (2021), qualitatively speaking we observe the same behaviors. For both $e_F$, the angle decreases as $e_G$ is reduced. Furthermore, for the same $e_G$, the angle is smaller for $e_F = 4$ than for $e_F = 2$. Given these results, we are confident that our implementation is correct and consistent with the one of Ringeisen et al. (2021).

[Figure]

Figure 1: Snapshot of shear deformation ($\%/day$) after one hour in an idealized uniaxial compression experiment.

2) "For VP experiments with a normal flow rule, Hutter and Losch (2020) compared the mean orientation of LKFs with the grid axis and could not find a clear correlation. For the SIREx project, Hutter et al. (2022) made the qualitative argument that the peak of the PDF at 90° is not caused by LKFs aligned with the grid as models with an unstructured grid also lead to a peak at 90°. Our analysis, involving the computation of the intersection angle for all the conjugate pairs, clearly indicates a tendency of simulated LKFs to be aligned with

[Figure]

Figure 2: Fracture angle as a function of $e_G$ in idealized uniaxial compression experiments for $e_F = 2$ (in red) and $e_F = 4$ (in blue), with the associated Arthur angles $\theta_A$ in the same color.

the grid." This is already the second aspect in which the results of this study seem to differ from existing work. To relate the observed behavior to a grid effect, a more in-depth investigation, as suggested in point 1, is required.

We agree that our presented results seem to contradict previous studies. At the same time, we want to highlight that this study is the first thorough quantitative analysis of the alignment of simulated LKFs with the grid axes and that the methodology differs from the ones in previous studies. Hutter and Losch (2020) binned all LKFs in 200x200 km boxes, computed the orientation distribution, and visually compared its mode(s) with the grid orientation. In this comparison, they could not find a distinct Arctic wide correlation between the mode and the orientation of the grid axes. Note that the orientation of an LKF contributing to the distribution within a box is defined as the mean orientation of segment of the LKF that is laying within the box. In doing so, the orientation of the entire LKF contributes to the analysis. Also, all LKFs, conjugate and non-conjugate ones as well as new and old LKFs, were averaged.

As already written in our manuscript, Hutter et al. (2022) only made the

qualitative argument that the peak at 90° is also found for models with unstructured grids and therefore likely not linked to the orientation of the grid axes, without further analyzing the orientation of LKFs in relation to the numerical grid used in the simulations. In our quantitative analysis, we only consider newly formed LKFs of conjugate faults. Also, the orientation is measured directly at the intersection point. In doing so, we omit the effect that advection and different type of deformations can have on the orientation of LKFs. In our specific analysis, we find that LKFs of conjugate faults tend to preferentially orient themselves along the numerical grid axes in a region close to the intersection point. In Fig. 1 and 2 of the manuscript this characteristic is also observed showing a preferred orientation at the intersecting points and bended LKFs with varying orientation away from the intersection point. Compared to previous studies, the presented analysis therefore offers new insights about the formation of fractures in VP-EVP models.

3) In line 351, the authors write: "As demonstrated in other studies, we find that the elastic-viscous-plastic (EVP) rheology with a normal flow rule (i.e., $e_G = e_F$) leads to a PDF of intersection angles (for conjugate pairs) with a peak at 90°. Using a non-normal flow rule with the elliptical yield curve, as introduced by Ringeisen et al. (2021), does not remedy this problem. Either with $e_G < e_F$ or $e_G > e_F$, the peak is still at 90°." In my opinion, this study cannot claim general validity of its findings, as it remains unclear what the cause of the observed failure is. In particular, it has not been convincingly ruled out that the observed effect may be due to details in the implementation. From my perspective, it is still possible that the effect observed by Ringeisen can also be reproduced in pan-Arctic simulations, provided that the setup is carefully implemented.

See our responses and modeling results above.

Minor comments:

1) Line 32: "It is expressed with the use of a plastic potential: the post-failure deformations are normal to the plastic potential." This sentence is extremely technical for an introduction and difficult to understand. Some readers may not know what it means that post-failure deformations are normal to the plastic potential. It should either be reworded and better explained or supplemented with a sketch or equation that clarifies the meaning.

We will provide a better introduction to the concept of yield curve and plastic potential in the revised manuscript. However, we don't think we need to provide a schematic as Ringeisen et al. (2019) and (2021) already include nice illustrations.

2) Lines 31–40: These are too technical and difficult to digest for an introduction. Please revise and invest more effort in explaining the technical terminology.

We are aware that our article is a bit niche and not accessible to all the researchers in the field. However, we refer to many published articles to support the introduction of the concepts and equations. This is our approach to make sure the article is not too long.

3) Move Section 2 to the appendix. The presentation of the concepts is currently very isolated, making it completely unclear how they relate to the equations that describe the rheology. For example, in line 60: Which system of equations are you referring to? Please specify clearly.

This is just a brief introduction about the plastic potential and yield curve. We prefer to keep section 2 here. People familiar with the standard VP rheology will easily recognize these equations. We will clarify line 60.

4) In addition to the equations describing the rheology or yield curve in the standard VP model and in Ringeisen's version, graphical representations would be helpful for better understanding the key parameters $e_F$, $e_G$, and $e$. In the current presentation, it is very difficult for readers from the Cryosphere community, who may not be familiar with all of Ringeisen's previous work, to follow the argument.

We refer to the papers of Ringeisen if the reader wants to have more details.

Line 65: Please explicitly state the constitutive equation.

We don't think this is needed as it is given in many papers (e.g. Hibler 1979).

Line 77: "With the non-normal flow rule, $\eta$ is now equal to $e_G^{-2}\zeta$." – Why? Please explain.

This is from the derivation of Ringeisen et al. (2021). We will modify the text to clarify this.

5) Line 136: Where does this choice come from? Why is $e_G$ set to $e_F/1.5$, $e_F$, or $1.5e_F$? Please justify.

The $e_F$ values were chosen to match the values used in previous studies (e.g. Dumont et al. 2009). Setting $e_G$ to either $e_F/1.5$, $e_F$, or $1.5e_F$ ensures that the experiments are consistent for the different $e_F$ and that a sufficiently large range of values are tested.

6) Line 155: "A kernel value of seven is used for the detection algorithm." – Why 7? What does it mean? Please explain.

We refer here to the kernel width used to determine the orientation of LKF at each grid point. A rotating 1-d kernel of size seven centered at a LKF pixel is used to determine the orientation perpendicular to the LKF by maximizing the standard deviation of total deformation within the kernel. This kernel size should be chosen to roughly represent the width of LKFs in the data. To make clearer what kernel size we are talking about, we moved this sentence two paragraphs below, where we discuss the morphological thinning and orientation of the LKF.

7) Lines 168–173: What is $\alpha$? What does "morphological thinning" mean? Please include a sketch to make this clearer.

The morphological thinning is associated with the LKF detection algorithm of Hutter et al. (2019). This part of the algorithm ensures that a detected pixel only belongs to a single LKF. We do not think we need to add anything as we refer to this paper which describes the method and includes a schematic.

8) Line 176: What do you mean by "conjugate fault lines"?

We will modify line 20 to state that conjugate pairs of LKFs (i.e., LKFs that form simultaneously under compressive stresses) are also referred to as conjugate fault lines.

9) Line 177: What exactly is the vorticity being analyzed?

We follow the work of Hutter et al. (2022) by using the simulated sea ice vorticity deformation to determine if the LKFs are conjugate fault lines.

10) Lines 177–182: This section is extremely technical and hard to understand. Either explanatory sketches should be added, or the content should be moved to the appendix.

We understand that our paper might not be accessible to all the sea ice community as it targets a small and specialized audience. In order to keep the paper concise we don't have the choice to refer to other scientific articles instead of explaining many complicated concepts related to sea ice rheology.

11) Figure 1a: To me, the structures shown look like numerical instabilities. It is possible that the modification proposed by Ringeisen increases the stiffness of the system of equations, making it harder to solve. While 900 iterations are considered sufficient in the standard EVP model, they may be inadequate for the modification investigated here. I therefore recommend conducting tests with smaller time steps and/or more iterations to check whether sharper structures can be resolved and whether the 900 EVP subcycles are indeed sufficient here.

It is indeed stated in Ringeisen et al. (2021) that the non-normal flow rule

leads to a more difficult numerical convergence. 900 EVP subcycles is already a lot (the default is 240 in CICE) and quite demanding in terms of computational time for the pan-Arctic experiments. Note, however, that in idealized experiments (as in Fig. 1) we have confirmed that the solution does not really change when using more than 900 subcycles (not shown).

12) Line 366: "Note that our conclusions remain the same whether the implicit approach (VP) in CICE or the explicit one (EVP) is used." There are numerous publications, including some by Lemieux, that show that ten Picard steps are insufficient to achieve a convergent solution. Therefore, it is possible that the observed effect is due to the inadequate accuracy of the approximation solutions in both setups (EVP and Picard).

Again, in terms of computational efficiency, it is too demanding to use too many Picard iterations in these realistic pan-Arctic simulations. Note again, that the simulated deformations using Picard are similar to the ones obtained with the EVP in our idealized experiments (not shown).

Jean-Francois Lemieux

---

## Author Comment (AC2)

**Initial response to reviewer 2 after first review**

26 May 2025

We thank Guillaume for his time, his suggestions and for his positive comments about our manuscript. Our responses below are in blue.

**Reviewer 2**:

This is a review of the manuscript entitled "Impact of non-normal flow rule on linear kinematic features in pan-Arctic ice-ocean simulations." The study discusses the effect of allowing the rheology used in the CICE large-scale sea ice model to deviate from the normal flow rule, which is normally used to determine the direction of deformations when sea ice undergoes plastic deformation. It is largely motivated by a previous study that showed that using a nonnormal flow rule resulted in more realistic linear kinematic features (LKFs) in a more idealized setup. In particular, it had been shown to significantly improve the model's ability to reproduce realistic intersection angles of conjugate faults. Here, the authors show that this is not the case. They discuss the potential reasons for this negative result compared to the more idealized study and suggest (with some supporting arguments) that these intersection angles may be largely constrained to follow the model grid axes, resulting in intersection angles often close to 90° in the case of the grid used here. Despite this negative result, they show that using a non-normal flow rule has other effects on LKF properties, which could make its use a convenient way to tune LKF characteristics in large-scale sea ice models.

The manuscript is well-written and clear. The scope is well-defined, and the analysis is sound and well-supported. The topic is perhaps a bit niche, but it aligns with previous studies published in the journal and is of interest to the sea ice modelling community targeted by The Cryosphere. The suggestion that the peak at 90deg partly results from the alignments of LKFs with the grid axes is sound. It may not be fully demonstrated here (as mentionned), but the authors have gathered enough elements to point to this cause in their results and discussion. The argument is strong enough to motivate the community interested in sea ice deformations to investigate the numerics instead of only focusing on the physics. Additionally, the fact that the non-normal flow rule can be "tweaked" to tune sea ice deformations properties is an interesting result for the sea ice modelling community. Therefore, I support the publication of this manuscript

in The Cryosphere. I do not have comments that I consider major, but I would recommend the minor revisions below:

We thank the reviewer for these positive comments.

General comments:

Given that one of the main results is that using a non-normal flow rule can be a convenient way to tune certain LKF properties independently, the authors may consider commenting on the implementation in CICE. Does it require substantial modification to the code? Does it significantly impact computational time?

This is a very good suggestion. We will comment in the revised manuscript on what is required to implement the non-normal flow rule. The implementation is actually incredibly simple. Indeed, only small modifications to the calculation of the shear ($\eta$) and bulk ($\zeta$) viscosities are required. With the standard VP rheology, $\eta = e^{-2}\zeta$ where $e$ is the ellipse ratio and $\zeta = P/2\Delta$ with $\Delta = [D_d^2 + e^{-2}(D_t^2 + D_s^2)^2]^{1/2}$. For the non-normal flow rule, $e^{-2}$ is simply replaced by $e_G^{-2}$ for computing $\eta$ while $e^{-2}$ in $\Delta$ is replaced by $e_F^2/e_G^4$. There is therefore no impact on the computational time.

I sometimes found the order in which the figures are referenced a bit confusing. For example, it appears that Figure 4 is referred to after Figures 5 and 7, and Figure 2 before Figure 1. This has little impact in the preprint, since all figures are at the end, but the authors may consider reordering the figures for the final version.

Thank you for pointing this out. We will reorder the figures in the revised manuscript.

Specific comments:

L213: I think "of" is missing (or something else)

This will be corrected.

L337/338: I find this sentence a bit unclear.

This will be rephrased in the revised manuscript.

L340/350: I find the ideas in this paragraph a bit hard to follow, I think it would benefit from a bit of rewriting (especially the sentence starting with "But clearly..")

We will carefully rephrase these sentences.

Jean-Francois Lemieux

---

## Author Response (AR1)

**Response to reviewers after first review**

**July 2, 2025**

We thank both reviewers for their time and for their useful comments and suggestions. Our responses below are in blue.

**Reviewer 1:**

This paper presents a detailed sensitivity study on the influence of a nonnormal flow rule in the viscous-plastic (VP) sea-ice rheology on the simulation of linear kinematic features (LKFs) in Arctic sea ice. The authors implement the concept of a plastic potential and non-normal flow rule introduced by Ringeisen et al. (2021) into the CICE sea ice model and conduct a comprehensive series of pan-Arctic simulations based on this implementation. However, the central result of the study is that the peak of the probability density function (PDF) of the simulated intersection angles between conjugate LKFs remains at 90°, whereas observational data show a peak around 45°. This finding is counterintuitive, as Ringeisen et al. (2021) demonstrated that using a non-normal flow rule enables the simulation of smaller intersection angles that align more closely with observations. This discrepancy raises a fundamental and scientifically important question: what is the underlying cause of this difference? Answering this question would be interesting and valuable for the community. Instead of systematically addressing this question, the authors limit themselves to a purely parametric sensitivity analysis. As a result, parts of the paper take on the character of a technical documentation of the non-normal flow rule implementation in CICE, without achieving the intended effect, namely more realistic intersection angles. In my view, the innovative contribution of this paper is therefore limited. Since the discrepancy between the idealized tests in Ringeisen et al. (2021) and the CICE implementation remains unresolved, the overall validity of the presented results is questionable. In my view, it is entirely possible that the peak in LKF intersection angles at 90 degrees arises from numerical details specific to the CICE implementation, and that a careful reproduction of the setup proposed by Ringeisen et al. could yield a peak at smaller angles. The authors' explanation, that this discrepancy is due to a grid effect, appears vague and insufficiently analyzed. Moreover, this hypothesis is not in line with previous studies, such as Hutter et al. (2022) and Hutter and Losch (2020).

We think our manuscript focused too much on the intersection angles and

did not convey enough that, beside the intersection angles, a non-normal flow rule is a very interesting and useful model capability. Indeed, the plastic potential allows to optimize the deformations while maintaining the same yield curve. Similarly, the yield curve can be modified to optimize simulated landfast ice and sea ice drift with little impact on the deformations. We have modified the abstract, some of the introduction and conclusions to emphasize these clear advantages of the plastic potential.

I therefore recommend that the manuscript place greater emphasis on the core scientific question: Why does the CICE implementation yield different results from the idealized tests by Ringeisen et al.? A targeted and transparent validation of the implementation is essential in this regard. For these reasons, I recommend a major revision.

We include a supplement to the revised version of the manuscript. This supplement describes the numerical experiments that were conducted to validate the implementation of the plastic potential and non-normal flow rule. In our initial response to the reviewers we implemented a similar experiment as the one of Ringeisen et al. (2021). For this final response, however, we spend some time implementing the Dirichlet condition for the velocity and were therefore able to implement the uni-axial compression experiment of Ringeisen et al. (2021). We also find that the intersection angles vary when changing  $e_F$  and  $e_G$  and that the simulated angles follow closely the theoretical Arthur angles. Moreover, we also confirm that the numerical solution is viscous-plastic (i.e. states of stress are inside or on the yield curve) even when  $e_G$  is not equal to  $e_F$ . We are confident that our implementation is correct and that it follows the one proposed by Ringeisen et al. (2021).

Giving a complete explanation why the non-normal flow rule with  $e_G < e_F$  does not lead to more realistic LKF intersection angles in pan-Arctic experiments is beyond the scope of this paper. In the mean time, we think that our study is sound and could inspire other researchers to further investigate this.

**Major comments:**

1) The authors write: "We don't think it is related to the use of a C-grid and a VP rheology by Ringeisen et al. (2021) as opposed to the B-grid with EVP dynamics used here." To validate this claim and, more importantly, to demonstrate that the implementation performs as intended, I suggest reproducing the idealized test case from Ringeisen et al. within CICE and first evaluating whether their results can be replicated. This would confirm that the basic implementation is correct and that differences in numerical treatment—such as C-grid vs. B-grid or the choice of solver—do not, as the authors suggest, contribute to the observed deviations in the results.

As stated in our response to reviewer 2, implementing the plastic potential

of Ringeisen et al. (2021) is incredibly simple; it requires minor modifications to the code. We have revised our code and confirm it is consistent with the plastic potential formulation of Ringeisen et al. (2021). The reviewer is invited to read the supplement section where we validated our implementation and overall reproduced the results of Ringeisen et al. (2021).

2) "For VP experiments with a normal flow rule, Hutter and Losch (2020) compared the mean orientation of LKFs with the grid axis and could not find a clear correlation. For the SIREx project, Hutter et al. (2022) made the qualitative argument that the peak of the PDF at 90° is not caused by LKFs aligned with the grid as models with an unstructured grid also lead to a peak at 90°. Our analysis, involving the computation of the intersection angle for all the conjugate pairs, clearly indicates a tendency of simulated LKFs to be aligned with the grid." This is already the second aspect in which the results of this study seem to differ from existing work. To relate the observed behavior to a grid effect, a more in-depth investigation, as suggested in point 1, is required.

We agree that our presented results seem to contradict previous studies. At the same time, we want to highlight that this study is the first thorough quantitative analysis of the alignment of simulated LKFs with the grid axes and that the methodology differs from the ones in previous studies. Hutter and Losch (2020) binned all LKFs in 200x200 km boxes, computed the orientation distribution, and visually compared its mode(s) with the grid orientation. In this comparison, they could not find a distinct Arctic wide correlation between the mode and the orientation of the grid axes. Note that the orientation of an LKF contributing to the distribution within a box is defined as the mean orientation of segment of the LKF that is laying within the box. In doing so, the orientation of the entire LKF contributes to the analysis. Also, all LKFs, conjugate and non-conjugate ones as well as new and old LKFs, were averaged.

As already written in our manuscript, Hutter et al. (2022) only made the qualitative argument that the peak at 90° is also found for models with unstructured grids and therefore likely not linked to the orientation of the grid axes, without further analyzing the orientation of LKFs in relation to the numerical grid used in the simulations. In our quantitative analysis, we only consider newly formed LKFs of conjugate pairs. Also, the orientation is measured directly at the intersection point. In doing so, we omit the effect that advection and different type of deformations can have on the orientation of LKFs. In our specific analysis, we find that LKFs of conjugate pairs tend to preferentially orient themselves along the numerical grid axes in a region close to the intersection point. In Fig. 1 and 2 of the manuscript this characteristic is also observed showing a preferred orientation at the intersecting points and bended LKFs with varying orientation away from the intersection point. Compared to previous studies, the presented analysis therefore offers new insights about the formation of fractures in VP-EVP models.

We added more details in the revised manuscript to explain why we draw a different conclusion than Hutter and Losch (2020) and Hutter et al. (2022).

3) In line 351, the authors write: "As demonstrated in other studies, we find that the elastic-viscous-plastic (EVP) rheology with a normal flow rule (i.e.,  $e_G = e_F$ ) leads to a PDF of intersection angles (for conjugate pairs) with a peak at 90°. Using a non-normal flow rule with the elliptical yield curve, as introduced by Ringeisen et al. (2021), does not remedy this problem. Either with  $e_G < e_F$  or  $e_G > e_F$ , the peak is still at 90°." In my opinion, this study cannot claim general validity of its findings, as it remains unclear what the cause of the observed failure is. In particular, it has not been convincingly ruled out that the observed effect may be due to details in the implementation. From my perspective, it is still possible that the effect observed by Ringeisen can also be reproduced in pan-Arctic simulations, provided that the setup is carefully implemented.

See our responses above and the supplement section of the revised manuscript.

**Minor comments:**

1) Line 32: "It is expressed with the use of a plastic potential: the post-failure deformations are normal to the plastic potential." This sentence is extremely technical for an introduction and difficult to understand. Some readers may not know what it means that post-failure deformations are normal to the plastic potential. It should either be reworded and better explained or supplemented with a sketch or equation that clarifies the meaning.

We refer to the papers of Ringeisen if the reader wants to have more details. We don't think we need to provide a schematic as Ringeisen et al. (2019) and (2021) already include nice illustrations.

2) Lines 31–40: These are too technical and difficult to digest for an introduction. Please revise and invest more effort in explaining the technical terminology.

We are aware that our article is a bit niche and not accessible to all the researchers in the field. However, we refer to many published articles to support the introduction of the concepts and equations. This is our approach to make sure the article is not too long.

3) Move Section 2 to the appendix. The presentation of the concepts is currently very isolated, making it completely unclear how they relate to the equations that describe the rheology. For example, in line 60: Which system of equations are you referring to? Please specify clearly.

This is just a brief introduction about the plastic potential and yield curve. We prefer to keep section 2 here. People familiar with the standard VP rheology will easily recognize these equations. We have clarified line 60 in the revised manuscript and now refer to the constitutive and momentum equations.

4) In addition to the equations describing the rheology or yield curve in the standard VP model and in Ringeisen's version, graphical representations would be helpful for better understanding the key parameters  $e_F$ ,  $e_G$ , and e. In the current presentation, it is very difficult for readers from the Cryosphere community, who may not be familiar with all of Ringeisen's previous work, to follow the argument.

We refer to the papers of Ringeisen if the reader wants to have more details.

Line 65: Please explicitly state the constitutive equation.

We don't think this is needed as it is given in many papers (e.g. Hibler 1979).

Line 77: "With the non-normal flow rule,  $\eta$  is now equal to  $e_G^{-2}\zeta$ ." – Why? Please explain.

This is from the derivation of Ringeisen et al. (2021). We have clarified the text about this.

5) Line 136: Where does this choice come from? Why is  $e_G$  set to  $e_F/1.5$ ,  $e_F$ , or  $1.5e_F$ ? Please justify.

The  $e_F$  values were chosen to match the values used in previous studies (e.g. Dumont et al. 2009). Setting  $e_G$  to either  $e_F/1.5$ ,  $e_F$ , or  $1.5e_F$  ensures that the experiments are consistent for the different  $e_F$  and that a sufficiently large range of values are tested.

6) Line 155: "A kernel value of seven is used for the detection algorithm." – Why 7? What does it mean? Please explain.

We refer here to the kernel width used to determine the orientation of LKF at each grid point. A rotating 1-d kernel of size seven centered at a LKF pixel is used to determine the orientation perpendicular to the LKF by maximizing the standard deviation of total deformation within the kernel. This kernel size should be chosen to roughly represent the width of LKFs in the data. To make clearer what kernel size we are talking about, we moved this sentence two paragraphs below, where we discuss the morphological thinning and orientation of the LKF.

7) Lines 168–173: What is  $\alpha$ ? What does "morphological thinning" mean? Please include a sketch to make this clearer.

The morphological thinning is associated with the LKF detection algorithm of Hutter et al. (2019). This part of the algorithm ensures that a detected pixel only belongs to a single LKF. We do not think we need to add anything as we refer to this paper which describes the method and includes a schematic.

8) Line 176: What do you mean by "conjugate fault lines"?

We don't use this expression in the revised manuscript. We now use the expression pairs of conjugate LKFs or just conjugate LKFs (i.e., LKFs that form simultaneously under compressive stresses).

9) Line 177: What exactly is the vorticity being analyzed?

We follow the work of Hutter et al. (2022) by using the simulated sea ice vorticity deformation to determine if the LKFs are pairs of conjugate LKFs.

10) Lines 177–182: This section is extremely technical and hard to understand. Either explanatory sketches should be added, or the content should be moved to the appendix.

We understand that our paper might not be accessible to all the sea ice community as it targets a small and specialized audience. In order to keep the paper concise we don't have the choice to refer to other scientific articles instead of explaining many complicated concepts related to sea ice rheology.

11) Figure 1a: To me, the structures shown look like numerical instabilities. It is possible that the modification proposed by Ringeisen increases the stiffness of the system of equations, making it harder to solve. While 900 iterations are considered sufficient in the standard EVP model, they may be inadequate for the modification investigated here. I therefore recommend conducting tests with smaller time steps and/or more iterations to check whether sharper structures can be resolved and whether the 900 EVP subcycles are indeed sufficient here.

It is indeed stated in Ringeisen et al. (2021) that the non-normal flow rule leads to a more difficult numerical convergence. This is also what we observe in our idealized numerical experiments described in the supplement section. For the pan-Arctic experiments, 900 EVP subcycles is already a lot (the default is 240 in CICE) and quite demanding in terms of computational time.

12) Line 366: "Note that our conclusions remain the same whether the implicit approach (VP) in CICE or the explicit one (EVP) is used." There are numerous publications, including some by Lemieux, that show that ten Picard steps are insufficient to achieve a convergent solution. Therefore, it is possible that the observed effect is due to the inadequate accuracy of the approximation solutions in both setups (EVP and Picard).

Again, in terms of computational efficiency, it is too demanding to use too many Picard iterations in these realistic pan-Arctic simulations. Note that the time step also has to be considered with respect to the numerical convergence. Our simulations with 10 Picard iterations and  $\Delta t$ =3 min have a better convergence than the ones of Lemieux and Tremblay (2009) with 10 Picard iterations and  $\Delta t$ =30 min. We don't think that the too wide simulated intersection angles are due to the numerical convergence.

**Reviewer 2:**

This is a review of the manuscript entitled "Impact of non-normal flow rule on linear kinematic features in pan-Arctic ice-ocean simulations." The study discusses the effect of allowing the rheology used in the CICE large-scale sea ice model to deviate from the normal flow rule, which is normally used to determine the direction of deformations when sea ice undergoes plastic deformation. It is largely motivated by a previous study that showed that using a nonnormal flow rule resulted in more realistic linear kinematic features (LKFs) in a more idealized setup. In particular, it had been shown to significantly improve the model's ability to reproduce realistic intersection angles of conjugate faults. Here, the authors show that this is not the case. They discuss the potential reasons for this negative result compared to the more idealized study and suggest (with some supporting arguments) that these intersection angles may be largely constrained to follow the model grid axes, resulting in intersection angles often close to 90° in the case of the grid used here. Despite this negative result, they show that using a non-normal flow rule has other effects on LKF properties, which could make its use a convenient way to tune LKF characteristics in large-scale sea ice models.

The manuscript is well-written and clear. The scope is well-defined, and the analysis is sound and well-supported. The topic is perhaps a bit niche, but it aligns with previous studies published in the journal and is of interest to the sea ice modelling community targeted by The Cryosphere. The suggestion that the peak at 90deg partly results from the alignments of LKFs with the grid axes is sound. It may not be fully demonstrated here (as mentionned), but the authors have gathered enough elements to point to this cause in their results and discussion. The argument is strong enough to motivate the community interested in sea ice deformations to investigate the numerics instead of only focusing on the physics. Additionally, the fact that the non-normal flow rule can be "tweaked" to tune sea ice deformations properties is an interesting result for the sea ice modelling community. Therefore, I support the publication of this manuscript in The Cryosphere. I do not have comments that I consider major, but I would recommend the minor revisions below:

We thank the reviewer for these positive comments.

General comments:

Given that one of the main results is that using a non-normal flow rule can be a convenient way to tune certain LKF properties independently, the authors may consider commenting on the implementation in CICE. Does it require substantial modification to the code? Does it significantly impact computational time?

This is a very good suggestion. The implementation is actually incredibly simple. Indeed, only small modifications to the calculation of the shear  $(\eta)$  and bulk  $(\zeta)$  viscosities are required. With the standard VP rheology,  $\eta = e^{-2}\zeta$  where e is the ellipse ratio and  $\zeta = P/2\Delta$  with  $\Delta = [D_d^2 + e^{-2}(D_t^2 + D_s^2)^2]^{1/2}$ . For the non-normal flow rule,  $e^{-2}$  is simply replaced by  $e_G^{-2}$  for computing  $\eta$  while  $e^{-2}$  in  $\Delta$  is replaced by  $e_F^2/e_G^4$ . There is therefore no impact on the computational time. We have added the following text at the end of section 2:

'As only small modifications to the definitions of  $\eta$  and  $\Delta$  are required, the implementation of the non-normal flow rule for the VP rheology with an elliptical yield curve is straightforward. There is also no impact on the computational efficiency.'

I sometimes found the order in which the figures are referenced a bit confusing. For example, it appears that Figure 4 is referred to after Figures 5 and 7, and Figure 2 before Figure 1. This has little impact in the preprint, since all figures are at the end, but the authors may consider reordering the figures for the final version.

We added a new figure showing the mask used for the analysis and the orientation of the grid axes. We have also moved Fig.4. It is now Fig.9 in the revised manuscript. All figures are now referred to in the correct order.

Specific comments:

L213: I think "of" is missing (or something else)

This has been corrected.

L337/338: I find this sentence a bit unclear.

We have rephrased this sentence.

L340/350: I find the ideas in this paragraph a bit hard to follow, I think it would benefit from a bit of rewriting (especially the sentence starting with "But clearly..")

We have rephrased and clarified the content of this paragraph in the revised manuscript.

Best regards, Guillaume Boutin

Thanks again to both reviewers.

Jean-Francois Lemieux

---

## Referee Report (RR1)

**Review of the revised manuscript**

The authors have shifted the emphasis from a strong focus on intersection angles to highlighting the broader benefits of introducing a non-normal flow rule. They argue that the plastic potential offers an interesting and useful modeling capability: it allows optimization of deformations while maintaining the same yield curve, and conversely, it permits adjustments of the yield curve to improve simulations of landfast ice and sea-ice drift with little impact on deformation patterns.

Overall, the manuscript has been improved compared to the previous version. The most important criticism, the missing validation of the implementation of the non-normal flow rule, has been convincingly addressed: the supplement now includes a uniaxial compression test (after Ringeisen et al. 2021). Methodological differences from earlier studies (Hutter & Losch 2020; Hutter et al. 2022) are also explained more clearly, and the quantitative analysis provides new insights into the relationship between the computational grid and LKF orientations.

Nevertheless, some issues remain. The central discrepancy, why pan-Arctic simulations continue to produce intersection angles peaking at 90° instead of ~45°, is acknowledged but not resolved. This limits the general validity of the conclusions. In addition, it is not correct to state that the intersection angles are "partly caused by the alignment of LKFs with the computational grid"; the results demonstrate correlation rather than a proven causal mechanism.

Finally, the manuscript remains highly technical in places. Requests for explanatory sketches or schematics were not implemented; instead, references are made to other papers. This does not resolve the issue, especially when the sections in question concern the authors' own newly developed methodology in python. It raises the question of why such material is included in the manuscript if, in its current form, it remains difficult for readers to follow. This should clearly be improved before publication. I recommend acceptance after minor revisions

**Minor comments (to be addressed before publication):**

- 1.20: Please add a citation to "for example to optimize landfast ice."
- 1.14: "We show that these frequent 90° angles are partly caused by the alignment of LKFs with the computational grid." What is shown is rather a correlation, not clearly a causal mechanism.
- 1.93: You are probably referring to the appendix here, please clarify.
- 1.280–288 (Numerical convergence): It would be helpful to test at least one run with substantially more Picard iterations to rule out that insufficient convergence affects the results.
- Fig. 1: The notation "i=100, j=100" in the caption is unclear, please explain what i and j stand for.
- p.8, 1.176–190: This section remains very technical. Since it introduces the new analysis tool, the description should be made more accessible to readers. A sketch showing an LKF, grid lines, and the angles under discussion would be very helpful.
- 1.309: Please explain what *N* min means.
- Fig. 12: Please also include the standard VP model to make the differences visible.
- Appendix A: Please add i/j axes to the figure.
- 1.423–434: A sketch would help illustrate the contour lines under discussion.

---

## Author Response (AR2)

**Response to reviewers after second review**

**September 16, 2025**

We thank the reviewer for his/her second review. Our responses below are in blue.

**Reviewer 1:**

The authors have shifted the emphasis from a strong focus on intersection angles to highlighting the broader benefits of introducing a non-normal flow rule. They argue that the plastic potential offers an interesting and useful modeling capability: it allows optimization of deformations while maintaining the same yield curve, and conversely, it permits adjustments of the yield curve to improve simulations of landfast ice and sea-ice drift with little impact on deformation patterns.

Overall, the manuscript has been improved compared to the previous version. The most important criticism, the missing validation of the implementation of the non-normal flow rule, has been convincingly addressed: the supplement now includes a uniaxial compression test (after Ringeisen et al. 2021). Methodological differences from earlier studies (Hutter and Losch 2020; Hutter et al. 2022) are also explained more clearly, and the quantitative analysis provides new insights into the relationship between the computational grid and LKF orientations.

Nevertheless, some issues remain. The central discrepancy, why pan-Arctic simulations continue to produce intersection angles peaking at 90° instead of 45°, is acknowledged but not resolved. This limits the general validity of the conclusions. In addition, it is not correct to state that the intersection angles are "partly caused by the alignment of LKFs with the computational grid"; the results demonstrate correlation rather than a proven causal mechanism.

Explaining why intersection angles are still at 90° instead of 45° is a difficult problem that would require a lot of work. This is discussed in the conclusion of the manuscript.

We agree about the correlation rather than a complete explanation of the mechanism. We have rephrased some sentences to reflect this.

Finally, the manuscript remains highly technical in places. Requests for explanatory sketches or schematics were not implemented; instead, references are made to other papers. This does not resolve the issue, especially when the sections in question concern the authors' own newly developed methodology in python. It raises the question of why such material is included in the manuscript if, in its current form, it remains difficult for readers to follow. This should clearly be improved before publication. I recommend acceptance after minor revisions.

We have clarified the text in Appendix A and added a figure to better explain the algorithm for calculating the width of LKFs. For the rest, in order to keep this manuscript relatively short, we prefer not to include similar schematics as the ones already presented in other articles.

**Minor comments:**

1) 1.20: Please add a citation to "for example to optimize landfast ice."

We have cited Lemieux et al. 2016.

2) l.14: "We show that these frequent 90° angles are partly caused by the alignment of LKFs with the computational grid." What is shown is rather a correlation, not clearly a causal mechanism.

We agree. It is written in the revised manuscript: 'Results suggest that these frequent 90° angles are partly caused by the alignment of LKFs with the computational grid.'

3) 1.93: You are probably referring to the appendix here, please clarify.

No, as stated we are referring to the supplement.

4) 1.280–288 (Numerical convergence): It would be helpful to test at least one run with substantially more Picard iterations to rule out that insufficient convergence affects the results.

The results with the implicit solver are really a small complement to the main results. We do not want to put more emphasis on this. Furthermore, simulations with 10 Picard iterations with a time step of 3 minutes are already quite demanding in terms of computational resources.

5) Fig. 1: The notation "i=100, j=100" in the caption is unclear, please explain what i and j stand for.

It is already written in the caption that the grid indices are i and j. We don't think more detail is needed.

6) p.8, l.176–190: This section remains very technical. Since it introduces the new analysis tool, the description should be made more accessible to readers. A sketch showing an LKF, grid lines, and the angles under discussion would be very helpful.

We have notably improved the text in Appendix A to better explain our new tool. We have also included a new schematic for explaining the calculation of the LKF width.

7) 1.309: Please explain what  $N_{min}$  means.

We have clarified this. It is now written in the revised manuscript: 'Increasing the minimum number of points required  $(N_{min})$  from 10 to 20 or 30 (i.e., analysing only longer and longer LKFs) does not change, qualitatively, our conclusions...'

8) Fig. 12: Please also include the standard VP model to make the differences visible.

Unfortunately, our simulations with the implicit solver (VP) do not cover the full period of the time series. Anyway, we would expect small differences and don't think this would be useful.

9) Appendix A: Please add i/j axes to the figure.

We added the x-y axes on Fig.(A1).

10) 1.423-434: A sketch would help illustrate the contour lines under discussion.

We agree that Appendix A needed to be reworked. We have therefore added Fig.(A2) that explains the procedure for calculating the half width. We have also modified and reorganized some of the text.

Thank you very much for your help in improving this article.

Jean-Francois Lemieux